# Phase separation of TPX2 enhances and spatially coordinates microtubule nucleation

Matthew R. King [1,2] & Sabine Petry [1*]

Phase separation of substrates and effectors is proposed to enhance biological reaction rates and efficiency. Targeting protein for Xklp2 (TPX2) is an effector of branching microtubule nucleation in spindles and functions with the substrate tubulin by an unknown mechanism. Here we show that TPX2 phase separates into a co-condensate with tubulin, which mediates microtubule nucleation in vitro and in isolated cytosol. TPX2-tubulin co-condensation preferentially occurs on pre-existing microtubules, the site of branching microtubule nucleation, at the endogenous and physiologically relevant concentration of TPX2. Truncation and chimera versions of TPX2 suggest that TPX2-tubulin co-condensation enhances the efficiency of TPX2-mediated branching microtubule nucleation. Finally, the known inhibitor of TPX2, the importin-$\alpha/\beta$ heterodimer, regulates TPX2 condensation in vitro and, consequently, branching microtubule nucleation activity in isolated cytosol. Our study demonstrates how regulated phase separation can simultaneously enhance reaction efficiency and spatially coordinate microtubule nucleation, which may facilitate rapid and accurate spindle formation.

---

[1] Department of Molecular Biology, Princeton University, Princeton, New Jersey 08544, USA. [2] Present address: Department of Biomedical Engineering, Washington University, Brauer Hall, One Brookings Drive, Saint Louis, Missouri 63130, USA. *email: spetry@princeton.edu

The microtubule (MT) cytoskeleton organizes the interior of the cell, determines cell shape, and segregates chromosomes. Underlying its timely and accurate formation are multiple MT nucleation pathways from various cellular locations that need to be turned on at the right cell cycle stage. Only few pathway-specific MT nucleation effectors are known and their molecular mechanisms remain poorly understood[1,2]. At the same time, pioneering in vitro studies have implicated a role for liquid–liquid phase separation (LLPS) of proteins in cytoskeletal assembly[3–7], but the exact physiological contribution remains unclear. More generally, many proteins have been shown to undergo LLPS in vitro, whereas functional roles of LLPS in cells remain to be discovered[8,9].

Branching MT nucleation is a recently identified pathway during which new MTs nucleate along the lattice of pre-existing ones[10]. It exponentially increases MT numbers while preserving their polarity and is critical for rapid and accurate spindle assembly in *Xenopus* and human cells, and in *Drosophila*[10–14]. Branching MT nucleation requires the universal MT nucleator module, consisting of the γ-Tubulin Ring Complex (γ-TuRC)[15,16] and its recently discovered co-factor XMAP215[17–19], as well as the protein complex augmin that directly recruits γ-TuRC along the length of a pre-existing MT[20]. Branching MT nucleation is initiated by the MT-associated protein TPX2[10], which has been proposed to both recruit augmin to MTs[21,22] and activate γ-TuRC via TPX2's C-terminal domain[23,24]. In vitro, TPX2 can directly generate MTs from tubulin via its N-terminal domain[25,26], but this domain is dispensable for MT nucleation in the cytosol[23,27]. Understanding how TPX2 stimulates MT nucleation could help pioneer how a specific MT nucleation pathway is turned on to build cellular MT structures such as the mitotic spindle.

Here we show that TPX2 undergoes phase separation to form a co-condensate with tubulin at its endogenous and physiologically relevant concentration in *Xenopus* egg cytosol. The co-condensation of TPX2 and tubulin occurs on MTs and thus helps to both specifically promote MT nucleation from pre-existing MTs and enhance branching MT nucleation rates in the cytosol. Lastly, importins regulate this process by inhibiting the formation of co-condensates. Collectively, these data provide a molecular mechanism for TPX2 function, which is not only critical to explain spindle assembly but also demonstrates that phase separation can spatially coordinate reactions and enhance reaction kinetics in a physiological context.

## Results

**TPX2 and tubulin co-condense in vitro and in the cytosol.** When characterizing TPX2, we noticed features of known phase-separating proteins: a disordered N-terminus and a more ordered C-terminus with potentially multivalent α-helical regions[23,28] (Fig. 1a). Using a standard phase-separation test[28,29], either green fluorescent protein (GFP)-tagged or untagged TPX2 in high-salt buffer (0.5 M KCl) was diluted to physiological salt levels (0.1 M), resulting in the formation of spherical condensates (Fig. 1b, see 1c for assay principle). These condensates fulfill several criteria of LLPS: they fuse, exhibit salt- and concentration-dependent condensation, and show fluorescence recovery that saturates over time (Supplementary Fig. 1a–c and Supplementary Movie 1).

We hypothesized that TPX2 may interact with tubulin dimers as a co-condensate, because the two do not interact as mono-dispersed, non-phase-separated proteins, yet form previously observed, but poorly understood higher-order structures alternately termed clusters, aggregates, or puncta. These structures nucleate MTs both in vitro[23,25–27] and in cells[30–32] via an unknown mechanism. When a non-phase-separated mixture of

TPX2 and tubulin at high salt is lowered to physiological salt levels (Fig. 1c), co-condensates with high partition coefficients are formed, indicating both proteins are enriched within the condensate relative to their surrounding buffer (Fig. 1d). Furthermore, TPX2 selectively co-condenses with tubulin but not with a protein of similar size and charge (Supplementary Fig. 1d). When phase separation occurs in a buffer that enables MT polymerization, TPX2-tubulin co-condensates generate MTs (Fig. 1e) to form an MT aster similar to those previously observed in vitro[23,25–27]. These data suggest that previous observations of TPX2 and tubulin clusters, aggregates, and puncta could have been manifestations of TPX2-tubulin co-condensation via LLPS.

To investigate whether TPX2 functionally co-condenses with tubulin in a physiological context, we pre-formed TPX2 condensates and overlaid them with meiotic *Xenopus laevis* egg cytosol containing soluble tubulin (Fig. 1f). Initially, TPX2 condensates selectively enriched tubulin from the isolated cytosol and then MTs grew from co-condensates to generate branched MT networks (Fig. 1g, h) that resembled previously observed TPX2-mediated branched MT networks[10,23]. The tubulin signal in the condensates diminished as they generated branched MT networks (Fig. 1h, i), but not as a result of photobleaching (Supplementary Fig. 2a). Interestingly, the physiological behavior of TPX2 to generate branched MT networks could only be observed with non-aged, liquid-like TPX2 condensates, but not with TPX2 condensates that had hardened after aging (Supplementary Figs. 1c and 2b–d). The latter condensates still enriched tubulin and generated either aster-like MT arrays (when aged 15 min; Supplementary Fig. 2c) or no visible MTs (when aged 30 min, Supplementary Fig. 2d). Collectively, our observations demonstrate that TPX2 and tubulin can undergo LLPS to form a co-condensate capable of generating MTs in vitro and in the cytosol. This characterization provides a mechanistic framework for how the two proteins may functionally interact.

**TPX2-tubulin preferentially co-condense on MTs.** As a major function of TPX2 is stimulating branching MT nucleation[10,23], we investigated whether TPX2-tubulin co-condensation contributes to this function. To observe TPX2 dynamics during branching MT nucleation, we added non-phase-separated GFP-TPX2 to *X. laevis* egg cytosol containing fluorescently labeled tubulin and end binding protein 1 (EB1) that tracks growing MT plus ends (Fig. 2a). We observed that TPX2 binds to the first emerging MT and then exclusively associates with the growing MT network (Fig. 2b and Supplementary Movie 2)[22]. Based on these results, we hypothesized that TPX2-tubulin co-condensates form preferentially on MTs, and that condensates in solution would only form in the absence of MTs. To test this, we prevented MT polymerization in cytosol via nocodazole and indeed observed that non-phase-separated TPX2 and tubulin added to the extract formed small, spherical, and highly mobile TPX2-tubulin co-condensates (Fig. 2c and Supplementary Movie 3). These are reminiscent of nocodazole-induced TPX2 puncta previously observed in cells[30,31]. Collectively, these data suggest that TPX2 and tubulin can form a co-condensate in the extract but TPX2 preferentially associates with MTs rather than form a condensate in solution.

To further investigate the conditions in which TPX2 and tubulin form co-condensates, we first mapped the phase boundary of TPX2 via partition coefficient (Supplementary Fig. 3a, b and see Fig. 1c for assay setup) and soluble pool measurements (Supplementary Fig. 3c). Interestingly, tubulin lowers the concentration at which TPX2 phase separates from ~200 nM to ~50 nM, which, compellingly, corresponds to the estimated concentration range for endogenous TPX2 in *X. laevis* egg cytosol of 25–100 nM

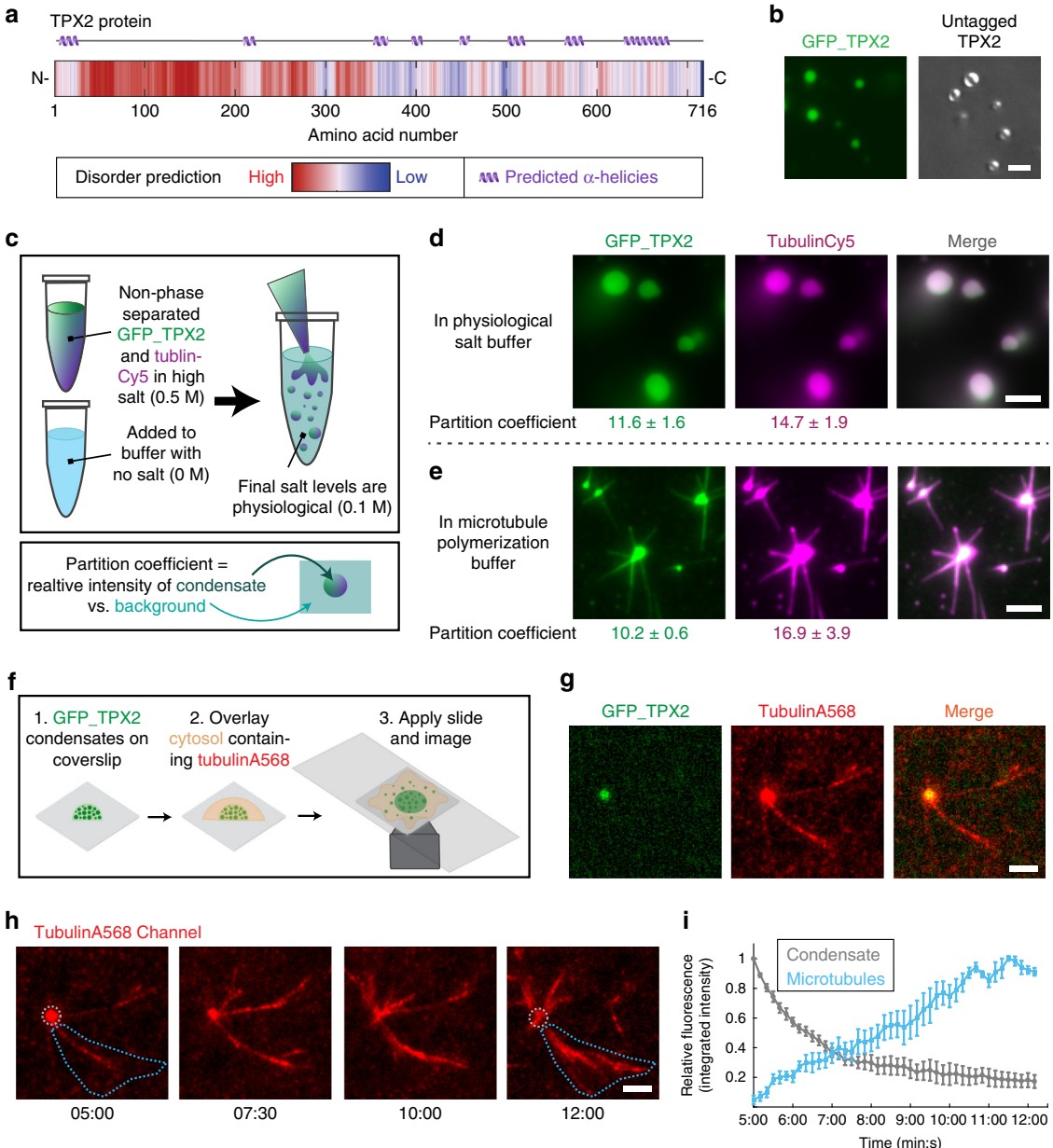

**Fig. 1 TPX2 forms a co-condensate with tubulin in vitro and in the cytosol.** All scale bars are 3 μm. **a** Secondary structure and intrinsic disorder predictions in TPX2. **b** Epifluorescent image of GFP-TPX2 (green) condensates (see Supplementary Movie 1) (left) and DIC image of untagged TPX2 condensates (right), both at a final concentration of 1 μM. Representative of six experimental replicates. **c** Schematic for assaying phase separation—TPX2 (with or without other proteins) is purified and maintained in a high-salt (0.5 M) buffer and this is transferred at 1:4 volume:volume into a no salt buffer to achieve physiological salt levels (0.1 M). **d** Epifluorescent image of GFP-TPX2 (green) condensates prepared with Cy5-labeled tubulin (magenta) (both at 4 μM) prepared as shown in **c** and imaged in a flow chamber (see Supplementary Fig. 1d for control). Representative of six experimental replicates. **e** TIRF image of TPX2-Tubulin co-condensates (green and magenta, 1 and 10 μM, respectively) prepared in MT polymerization buffer in a flow chamber, 18 min after reaction started. Representative of three experimental replicates. Partition coefficients for **d** and **e** are mean values (points) with ±1 SD as error bars from 225 and 170 condensates, respectively. **f** Experimental setup for **g**—pre-formed TPX2 condensates are overlaid with *Xenopus* egg cytosol containing fluorescent tubulin. **g** Oblique-TIRF microscopy of GFP-TPX2 (green) and tubulin (Alexa568-labeled—red) taken 5 min after reaction started (minutes: seconds). **h** In the same experiment as shown in Fig. 1g, the tubulin channel imaged over time (minutes:seconds) and depicted. Data representative of three experimental replicates. **i** Quantification of integrated tubulin signal from indicated areas corresponding to initial condensates (gray) and MT fan structures (blue). Mean values shown as circles with ±1 SD shown as error bars from $n = 5$ technical replicates. Source data are provided as a Source Data file.

(Supplementary Fig. 3a–c)[22,33]. Surprisingly, TPX2 localizes to MTs at the much lower concentration of 1 nM (Fig. 2d, e), which is 200-fold and 50-fold lower than the phase boundary for TPX2 and TPX2-tubulin condensates, respectively, in solution (Supplementary Fig. 3a–c)[25]. These data, too, suggest that TPX2 prefers to bind to MTs rather than associate with itself or tubulin in solution.

Next, we sought to determine whether TPX2 indeed forms a liquid-like co-condensate with tubulin on pre-existing MTs (see Fig. 2h for assay principle). First, we observed that TPX2 specifically recruits soluble tubulin along the length of MTs (Supplementary Fig. 3d). Compellingly, this occurs at the same concentration of 50 nM TPX2, which corresponds to the phase boundary of TPX2-tubulin co-condensation in solution (Fig. 2d, g

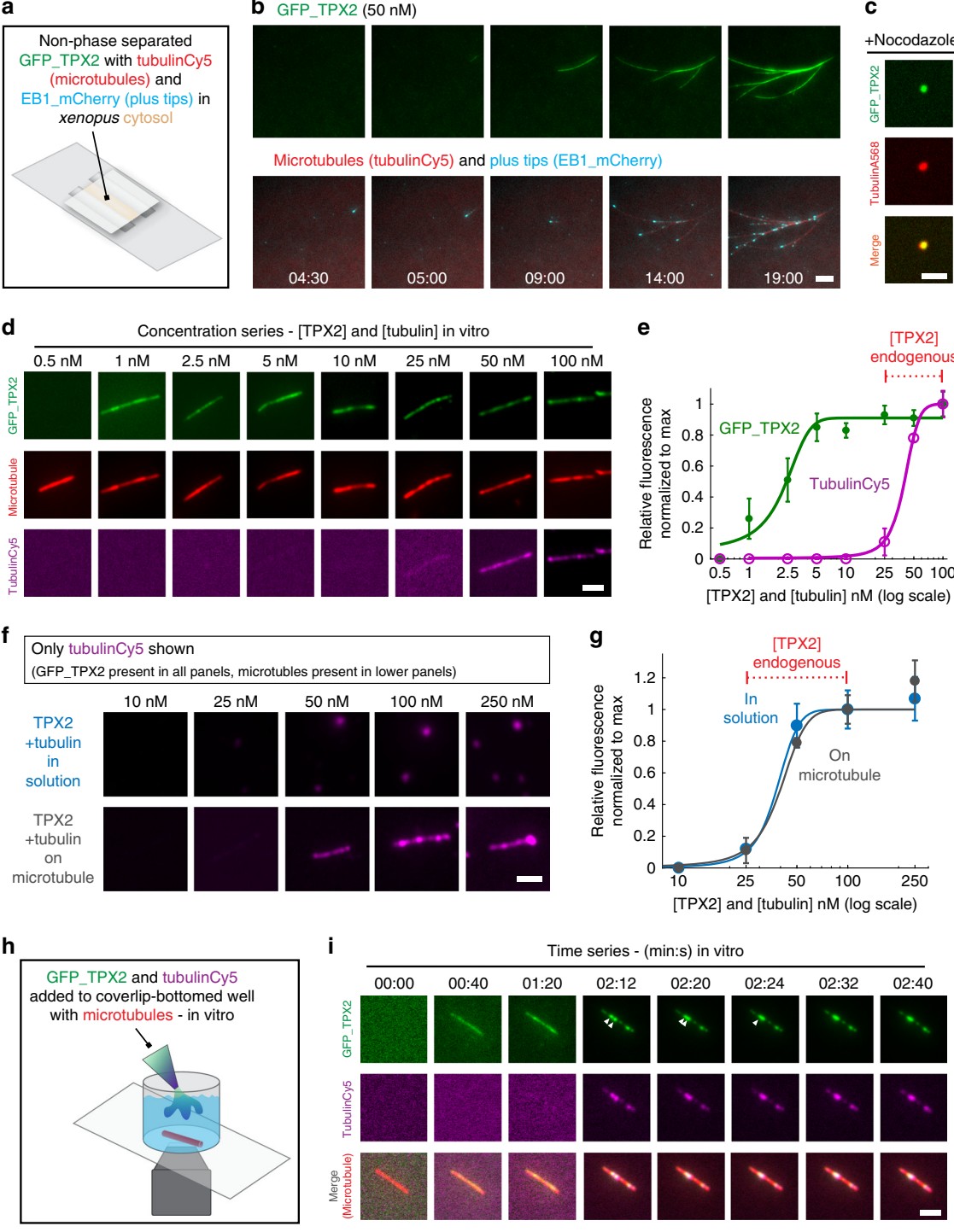

and Supplementary Fig. 3e). Moreover, at concentrations ≥50 nM, TPX2 and tubulin formed a puncta along the MT, which, importantly, are similarly sized to condensates in solution (Fig. 2d–g, i and Supplementary Fig. 3d). Lastly, we note that these condensates were generated from a lower concentration of TPX2 and are correspondingly smaller than those shown in Fig. 1, which is consistent with other phase-separating proteins[4,5,34].

By imaging TPX2 and tubulin recruitment to MTs live (Fig. 2h), we observed that they exhibit liquid-like co-condensate features. Initially, TPX2 uniformly coated the MT lattice and then beaded up into apparent condensates that fused together and

eventually became distributed along the MT lattice (Fig. 2i) (Supplementary Movie 4). This process can potentially be described by Rayleigh instability of a fluid[35]. We also sought to test whether depolymerizing TPX2-coated MTs leads to TPX2 condensate formation in solution; however, TPX2-coated MTs were resistant to depolymerization by taxol washout (Supplementary Fig. 4a), consistent with TPX2's known role in stabilizing MTs[36,37]. Fluorescence recovery after photobleaching (FRAP) of TPX2 on MTs revealed that TPX2 has similar recovery kinetics on MTs as it does as a condensate in solution (Supplementary Fig. 4b, c). Moreover, FRAP of tubulin within the TPX2-tubulin co-condensate on the MT exhibits similar recovery kinetics to

**Fig. 2 TPX2 preferentially co-condenses with tubulin on MTs. a** Experimental setup for **b** and **c**: non-phase-separated GFP-TPX2 (50 nM) and Alexa568-labeled tubulin mixed with *Xenopus* meiotic egg cytosol and imaged via oblique-TIRF microscopy. **b** GFP-TPX2 (green) localization to growing MT network imaged over time (minutes:seconds) in the cytosol (see Supplementary Movie 2), representative of three experimental replicates. Microtubules are labeled red and growing plus-tips are blue. All scale bars are 3 μm, unless indicated. **c** GFP-TPX2 (green) - and Alexa568-labeled tubulin (red) colocalization in the cytosol treated with Nocodazole to prevent MT polymerization, representative of five experimental replicates. Images taken 10 min into reaction (see Supplementary Movie 3). Scale bar is 3 μm. TPX2 was 50 nM and tubulin was 2 μM. **d** TIRF images (contrast-optimized), representative of four experimental replicates, and **e** quantification of GFP-TPX2 (green) and Cy5-labeled tubulin (magenta) (at indicated concentrations, both proteins equimolar) localized to pre-formed microtubules (Alexa568-labeled, GMPCPP stabilized—red) and spun down onto coverslips, fixed, and imaged; scale bar, 3 μm. **f** Oblique-TIRF images of only Cy5-labeled tubulin (magenta) condensed with GFP-TPX2 (not shown) either in solution (top panel) or on a pre-formed MT (lower panel—MT not shown) at concentrations shown. Images displayed at matched brightness and contrast. Enhanced contrast of 10 nM and 25 nM images shown is shown in the box. Scale bar, 2 μm. **g** Quantification of relative tubulin signal from TPX2-tubulin co-condensates in solution (blue curve) and on microtubules (purple curve) at concentrations shown in **e**. Mean (points) and SEM (error bars) of three replicate experiments (error bars) shown, n of microtubules per condition are: 145, 216, 114, 217, 243, 335, 309, and 309 for 1, 2.5, 5, 10, 25, 50, 100, and 250 nM [TPX2/TB], respectively; n of condensates per condition are: 280, 225, 266, and 282 for 10, 25, 50, 100, and 250 nM [TPX2/TB], respectively. Endogenous concentration range of TPX2 (q = 25–100 nM) indicated. **h** Schematic of experimental setup—live observation of TPX2 (green) and soluble tubulin (magenta) localization to a stable microtubule (red) attached to a coverslip-bottomed well. **i** TIRF images of a time series of GFP-TPX2 (green) and soluble Cy5-labeled tubulin (magenta) localization to a GMPCPP-stabilized Alexa568-labeled microtubule seed (red) (see Supplementary Movie 4). Images are contrast enhanced to show early non-association events and are representative of three experimental replicates. Arrowheads indicate fusion of two condensates. Time in minutes:seconds, 00:00 corresponds to addition of TPX2 into the well. Scale bar, 2 μm. Source data are provided as a Source Data file.

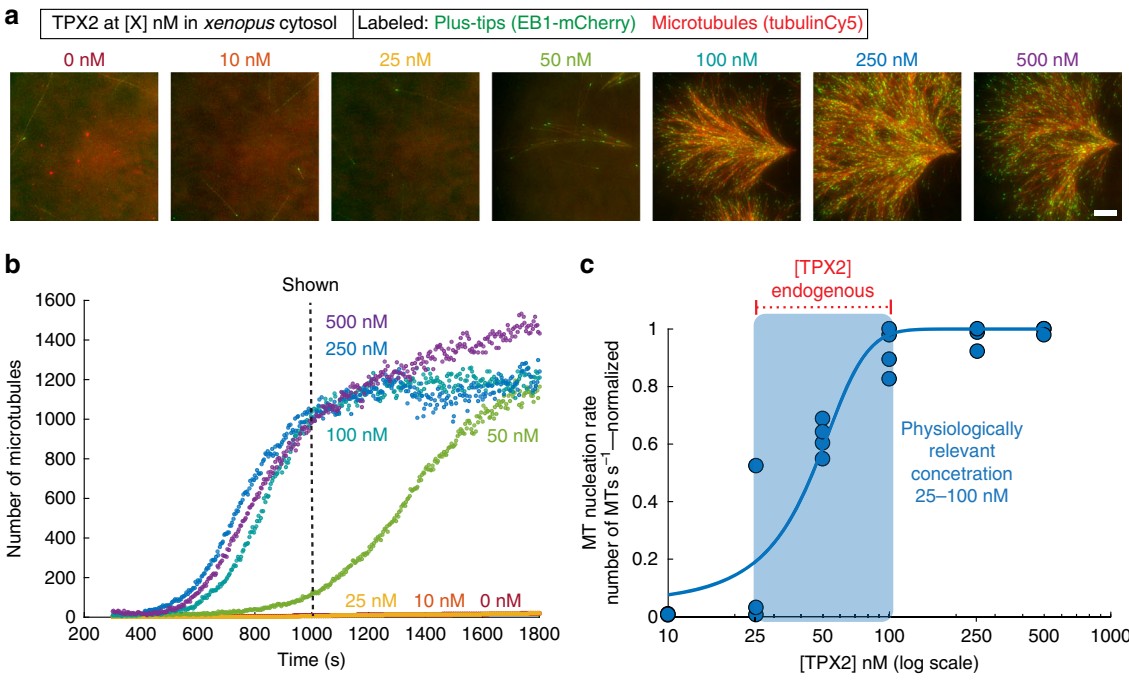

**Fig. 3 TPX2-tubulin co-condense at the physiologically relevant concentration of TPX2. a** TIRF images and **b** quantification of TPX2-mediated branching MT nucleation in *Xenopus* meiotic cytosol at indicated concentrations of TPX2. Shown images were taken at 1000 s (indicated). Cy5-labeled tubulin (red) and mCherry-EB1 (green) highlight microtubules and growing microtubule plus ends, respectively. Scale bar, 10 μm. See also Supplementary Movie 5. **c** Rate of MT nucleation as a function of TPX2 concentration for four independent replicates of data shown in **a** and **b**. Source data are provided as a Source Data file. See also Table 1. Rates normalized to maximum rate within an experiment. Line of best fit shown and approximate physiologically relevant range (25–100 nM) highlighted. Endogenous concentration range of TPX2 (30–100 nM) indicated (also in Fig. 2e).

TPX2 (Supplementary Fig. 4d–f). Therefore, in addition to being morphologically similar (Fig. 2d–g and Supplementary Fig. 3a), TPX2-tubulin co-condensates exhibit a similar liquid-like nature on MTs (Supplementary Fig. 4b, c) as in solution (Supplementary Fig. 1c). Lastly, our time-course experiments showed that soluble tubulin was only recruited to the MT lattice when TPX2 started to bead up into apparent condensates (Fig. 2i and Supplementary Movie 4), further suggesting that these proteins exclusively interact as co-condensates. Collectively, these data suggest that TPX2 preferentially associates with tubulin as a LLPS co-condensate on MTs rather than in solution.

**TPX2 condenses with tubulin at physiological concentration.** To determine whether the preferential co-condensation of TPX2 and tubulin on MTs is important for TPX2 function, we first determined the physiologically relevant concentration of TPX2 for mediating branching MT nucleation. Endogenous TPX2 was removed by immunodepletion from the cytosol and non-phase-separated, recombinant TPX2 was added at various concentrations. The resulting MT nucleation kinetics of branched MT networks were measured (Fig. 3a, b and Supplementary Movie 5) and plotted as a function of TPX2 concentration (Fig. 3c). TPX2 increases the rate of MT nucleation roughly 100-fold (Table 1) in

**Table 1 MT nucleation rates of full-length TPX2.**

| [TPX2] (nM) | Fold change in rate (relative to 0 nM) | Range (of replicates) |
|---|---|---|
| 0 | 1 | 1 |
| 10 | 1 | 0.9-1.2 |
| 25 | 20 | 1-56 |
| 50 | 81 | 60-97 |
| 100 | 130 | 90-165 |
| 250 | 133 | 93-174 |
| 500 | 137 | 100-178 |

Fold change in rates of MT nucleation relative to no TPX2 added (0 nM). Rates were normalized to a rate of 1 in the 0 nM condition within each experimental set and these values were averaged across replicate sets (second column). Range of fold-change values shown of the experimental replicate sets (three or four total measurements were obtained per concentration among four replicate sets). Rates for each concentration are also displayed as individual points in Fig. 2f. Source data are provided as a Source Data file

a switch-like manner within a concentration range of 25–100 nM (Fig. 3c). Strikingly, this concentration range precisely matches endogenous TPX2 levels and, most importantly, the phase boundary of TPX2-tubulin co-condensates, which are expected to be exclusively localized along MTs at this concentration. Collectively, these observations suggest that TPX2-tubulin co-condensation on a MT could underlie TPX2's switch-like activation of branching MT nucleation, as well as spatially bias MT nucleation to occur exclusively from pre-existing MTs.

**Disordered N-terminus enhances condensation and function**. To determine how TPX2-tubulin co-condensation contributes to branching MT nucleation, we first tested various truncations of TPX2. Although co-condensation was not completely abrogated in any truncation, the disordered N-terminal constructs phase separated at much lower concentrations and had greater tubulin partition coefficients than C-terminal constructs (Fig. 4a–c and Supplementary Fig. 5a–d). Although the N-terminal 1–480aa region drives the majority of TPX2 phase transition and tubulin co-condensation, it alone does not stimulate branching MT nucleation, whereas the minimal fragment that retains branching activity is TPX2's C-terminal 480–716aa (Fig. 4d, Supplementary Fig. 5e, f and Supplementary Movie 6)[23]. Yet, although not functional on its own, within the context of the full-length protein, the N-terminal region of TPX2 enhanced the efficiency of MT nucleation about tenfold (Fig. 4e).

**Co-condensation underlies efficient branching MT nucleation**. Next, we researched the mechanism by which TPX2 stimulates branching MT nucleation and the precise role of TPX2-tubulin co-condensation in this process. To do this, we replaced the N-terminal 1–480aa with various heterologous regions to generate TPX2 chimeras with distinct functionalities (Fig. 5a). We tested the ability of these chimeras to co-condense with tubulin (Fig. 5b) and promote branching MT nucleation (Fig. 5c). Due to the presence of the C-terminal minimal fragment (CT_480-716)[23], all of the chimeras were still capable of eliciting branching MT nucleation with reduced MT nucleation efficiency relative to full length.

First, the intrinsically disordered N-terminal region of fused in sarcoma (FUS) was used to replace TPX2's N-terminal part, as it readily phase separates[38], but is not expected to associate with tubulin owing to its negative pI (IDR_NoTB; Fig. 5a). Conversely, TOG domains 1 and 2 from XMAP215 were used, because they are well structured and are not expected to contribute to phase separation, but validated to functionally interact with tubulin[19,39,40] (NoIDR_TB; Fig. 5a). Neither the chimera containing an N-terminal region that confers only phase

separation but no additional tubulin condensation (IDR_NoTB), nor one that associates with tubulin via two structured TOG domains but does not improve phase separation (NoIDR_TB), changed the reduced MT nucleation efficiency of CT-TPX2 (Fig. 5b–d, Supplementary Fig. 6a–d, and Supplementary Movie 7). Interestingly, the NoIDR_TB chimera only showed a slight improvement in tubulin co-partitioning relative to the IDR_NoTB in vitro; however, the TOG domains used here have previously been validated to contribute the majority of the functional interaction with tubulin in *Xenopus* egg cytosol[19,40].

Next, the intrinsically disordered region of the MT-binding protein BuGZ was used[5]. This BuGZ domain phase separates and has a similar pI to the endogenous TPX2 N-terminus (IDR_TB), but does not independently promote branching MT nucleation (Supplementary Fig. 6e). Remarkably, this chimera recapitulated the TPX2-tubulin co-condensation property of full-length TPX2 (Fig. 5b and Supplementary Fig. 6f) and, most importantly, rescued the ~10-fold loss in MT nucleation efficiency in the cytosol that arises without the endogenous N-terminal 1–480aa region (Fig. 5c, d, Supplementary Fig. 7g, and Supplementary Movie 8).

Within TPX2, both aromatic and charged residues are highly conserved (Supplementary Fig. 7a). Aromatic residues have been implicated in tubulin condensation of the MT-associated protein BuGZ[5]; however, their removal from TPX2 does not alter tubulin co-condensation (Supplementary Fig. 7b–d). We hypothesized that the abundant and conserved charged residues may mediate tubulin co-condensation and efficiency of branching MT nucleation. To directly test this, we created a chimera using a synthetic peptide (Syn_Pos) that replicated both the positive charge and intrinsic disorder of the endogenous TPX2 region (Fig. 5a). Again, this chimera fully rescued TPX2-tubulin co-condensation and the MT nucleation efficiency of full-length TPX2 (Fig. 5b, c and Supplementary Fig. 7e, f). Critically, the gain-of-function chimeras (IDR_TB-CT-TPX2 and Syn_Pos-CT-TPX2) strongly implicate tubulin co-condensation as the underlying property that enables the physiological role of TPX2, namely, to mediate branching MT nucleation in its endogenous range (Fig. 5d).

**Importins α/β inhibit TPX2 condensation and activity**. Given the importance of TPX2-TB co-condensation to branching MT nucleation, we next sought to determine whether the property was functionally regulated. The importin-α/β heterodimer inhibits TPX2 until it is released by RanGTP at the onset of mitosis[41]. As RanGTP exists as a gradient emanating from chromosomes, the effective concentration of importins-α/β is low near chromosomes but increases further away into the spindle[41]. Correspondingly, we find that importins-α/β reduce both TPX2-tubulin co-condensation in vitro and TPX2-mediated MT nucleation in cytosol in a concentration-dependent manner: at twofold, excess importins-α/β and higher, both are abrogated (Fig. 6a–e and Supplementary Movie 9). It was previously determined that the gradient of active importins-α/β results in a sharp boundary of MT nucleation a certain distance away from chromosomes[12] and this distance can be modulated by altering global levels of TPX2[42]. Our findings indicate that this sharp MT nucleation boundary could be due to TPX2-tubulin co-condensation and its threshold regulation by importins-α/β.

**Discussion**
Our work elucidates how TPX2 and tubulin interact on a pre-existing MT, which provides a potential mechanism for TPX2's ability to specifically stimulate branching MT nucleation. More broadly, our findings serve as a proof-of-concept that phase

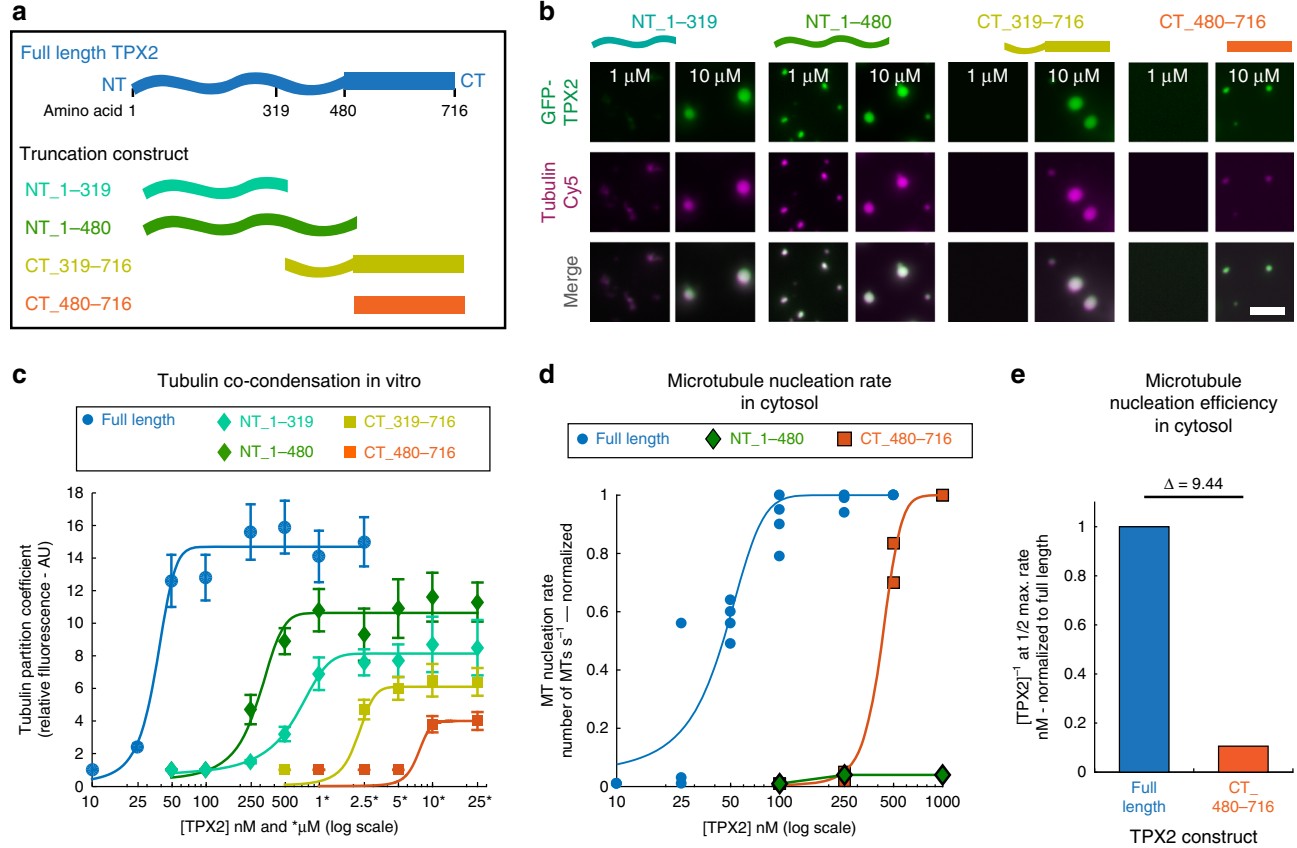

**Fig. 4 TPX2 N-terminus enhances condensation and MT nucleation efficiency. a** Schematic of full-length TPX2 and various N-terminal and C-terminal truncation constructs used. **b** Epifluorescent images of GFP-TPX2 (green) and Cy5-labeled tubulin (magenta) co-condensates at 1 μM and 10 μM (equimolar concentration) for indicated TPX2 truncation construct. Representative of three experimental replicates. **c** Quantification of relative tubulin signal in co-condensate (partition coefficient) as a function of TPX2 concentration. Mean values (points) with ± 1 SD as error bars from a representative experiment are plotted and a line was fit; *n* of condensates per condition are: NT_1-319—406, 572, 278, 217, 292, 276, and 268 for 250, 500 nM, 1, 2.5, 5, 10, and 25 μM [TPX2/TB], respectively; NT_1-480—254, 49, 256, 286, 283, 273, 271, and 198 for 100, 250, 500 nM, 1, 2.5, 5, 10, and 25 μM [TPX2/TB], respectively; CT_319-716—275, 323, 296, and 297 for 2.5, 5, 10, and 25 μM [TPX2/TB], respectively; CT_480-716—327 and 253 for 10 and 25 μM [TPX2/TB], respectively. See also Supplementary Fig. 5a–d for partition coefficient measures of GFP-TPX2 signal. **d** Rate of MT nucleation as a function of TPX2 concentration for indicated constructs. Full-length data previously shown (Fig. 3c). Rates for each concentration of a given construct are normalized to the maximum rate of that construct (absolute maximum rates between all constructs are within a twofold range). Lines of best fit shown. See also Supplementary Fig. 5e, f for individual rate curves and Supplementary Movie 6. **e** Different efficiencies of MT nucleation for the full-length and CT_480-716 TPX2 construct. Efficiency values are the inverse of the TPX2 concentration at which half the maximum rate of MT nucleation is achieved ([TPX2]$^{-1}$ at Rate$_{1/2Max}$); efficiency values were normalized to full length (efficiency of 1). The difference (fold change) in efficiencies is shown (Δ). Source data are provided as a Source Data file.

separation can spatially coordinate and enhance reactions in a physiological context.

Our findings suggest that TPX2 and tubulin interact via LLPS driven by electrostatic residues located primarily in the N-terminal 1–480aa of TPX2. TPX2-tubulin co-condensates preferentially form on pre-existing MTs and at a threshold concentration of TPX2 (~50 nM), which corresponds to its endogenous and physiologically relevant concentration. Collectively, our data suggest a model in which TPX2 pools tubulin via phase separation to create a local reservoir on a pre-existing MT (Fig. 7, right box), and that this may be necessary to efficiently promote branching MT nucleation in an all-or-none manner (Fig. 7, graph). Indeed, recent reconstitution experiments have shown that sites of condensation may serve as branch points for new MTs[21].

Our data indicate that dynamic, unaged, liquid-like TPX2 condensates most readily reproduce the physiologically observed branched MT networks; however, TPX2 condensates rapidly lose dynamics with age in vitro and it remains to be investigated whether this coincides with a hardening process and potentially more ordering of TPX2. Whether this aging occurs in vivo and whether it has functional consequences remain important open questions. Interestingly, intermediately aged TPX2 condensates generate an aster of MTs in *Xenopus* cytosol reminiscent of the centrosome. Notably, TPX2 accumulates at the centrosome as mitosis progresses and centrosomally localized TPX2 is less dynamic than TPX2 near chromosomes[30]. An important future research question is whether TPX2 dynamics change in a spatiotemporal manner commiserate with the MT nucleation needs of the spindle.

Branching MT nucleation requires the C-terminal 480–716aa of TPX2, which may function through an interaction with γ-TuRC[22–24] and its co-nucleation factor XMAP215[7,17–19,25]. Moreover, branching MT nucleation requires direct localization of γ-TuRC to pre-existing MTs via the protein complex augmin[20]. Interestingly, it was recently discovered that TPX2 must

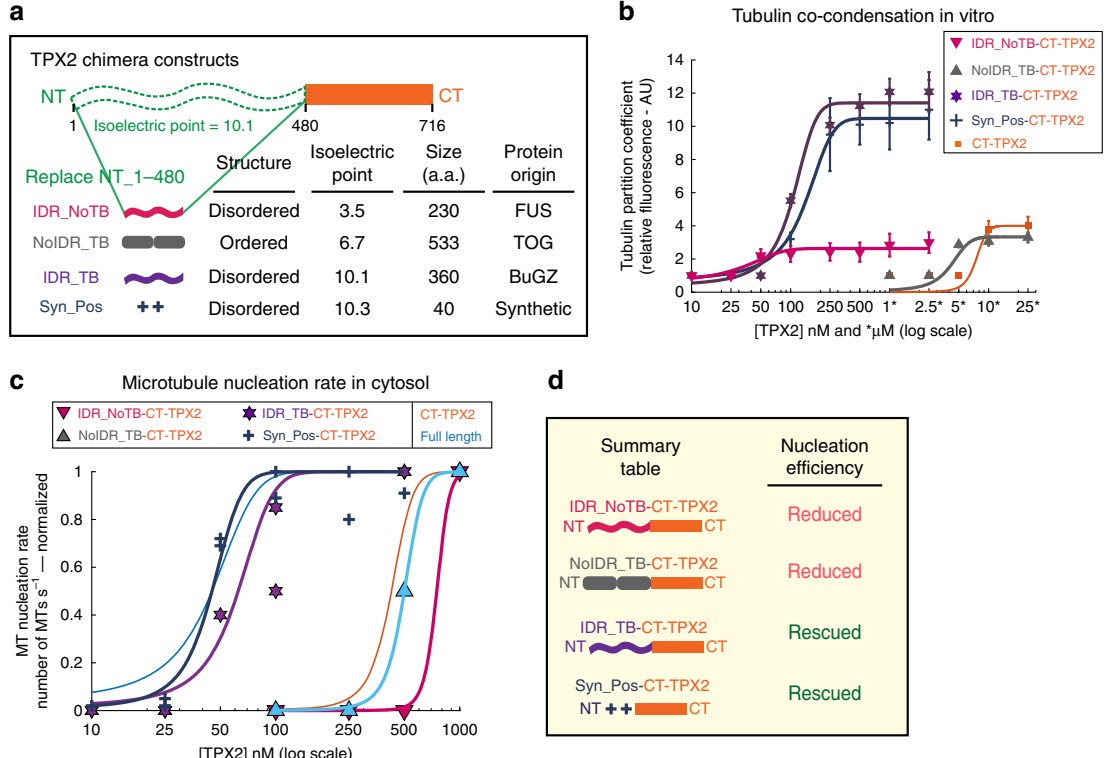

**Fig. 5 TPX2-tubulin co-condensation underlies efficient branching MT nucleation. a** Schematic of chimera design: the endogenous N-terminal 1–480aa of TPX2 are replaced with exogenous domains containing distinct features shown in table. **b** Quantification of tubulin partition coefficient as a function of TPX2 concentration. For partition coefficient graphs, mean values (points) with ±1 SD as error bars from a representative experiment are plotted and a line was fit; $n$ of condensates per condition are: IDR_NoTB—111, 80, 142, 99, 198, and 198 for 50, 100, 250, 500 nM, 1, and 2.5 μM [TPX2/TB], respectively; NoIDR_TB—198, 136, and 86 for 5, 10, and 25 μM [TPX2/TB], respectively; IDR_TB—87, 122, 153, 198, and 198 for 100, 250, 500 nM, 1, and 2.5 μM [TPX2/TB], respectively; Syn_Pos—326, 302, 100, 261, 270, and 323 for 50, 100, 250, 500 nM, 1 and 2.5 μM [TPX2/TB], respectively. See also Supplementary Figs. 6a, c, f and 7e for partition coefficients of TPX2. **c** Rate of MT nucleation as a function of TPX2 concentration for indicated constructs. Rates for each concentration of a given construct are normalized to the maximum rate of that construct (absolute maximum rates between all constructs are in a ±2× range). Lines of best fit shown. Data for full length (Figs. 2e and 3d) and CT_TPX2 (Fig. 4d) were previously shown. See also Supplementary Fig. 6b, d, e, g and 7f for individual rate curves and Supplementary Movies S7 and S8. Source data are provided as a Source Data file. **d** Summary table of results. MT nucleation efficiencies are relative to full-length TPX2.

first be deposited on MTs for augmin to be recruited[21,22]. Furthermore, TPX2 interacts with the tetrameric kinesin Eg5[30,43], which was recently demonstrated to promote MT nucleation in vitro[44]. Thus, TPX2 likely initiates branching MT nucleation by scaffolding not only tubulin, primarily through its N-terminal region, but also other essential nucleation factors, primarily through its C-terminal region. The data shown here also suggest that the phase separation property of TPX2 may enable it to scaffold tubulin for efficient MT nucleation. Importantly, similar mechanisms of phase separation-mediated scaffolding and enhancement of efficiency have recently been proposed for *Caenorhabditis elegans* centrosomes and mammalian oocytes spindle poles, suggesting that this may represent a general principle of MT organizing centers[7,34,45].

Loss of TPX2 results in reduction of spindle MT mass and mitotic delay[46], which is early embryonic lethal in mice[47] and causes apoptosis in proliferative somatic cells[48]. TPX2 overexpression leads to aberrant additional puncta of MT nucleation[30,32] that our data suggests correspond to TPX2 condensates. Furthermore, TPX2 overexpression is observed in 27% of all cancer types[46] and the degree of TPX2 overexpression tightly correlates with disease lethality[49,50]. Collectively, these studies demonstrate that maintaining appropriate TPX2 levels is essential for healthy cell division. Our work provides a potential mechanistic framework to understand the relationship between TPX2 levels and its associated MT nucleation activity, which could educate future therapeutic efforts aimed at modulating cell division.

Almost two decades ago, TPX2 was identified as the major downstream factor of RanGTP that is required for MT generation from chromatin, but the molecular mechanism of its regulation has remained unclear[46,51]. It was recently proposed the importins-α/β could directly block two MT-binding site on TPX2[36,52], but TPX2 can also localize to MTs via other sites[23,27] and importins-α/β do not inhibit TPX2 localization to MTs[53]. Our observation of complete inhibition of both TPX2-tubulin co-condensation and TPX2-mediated MT nucleation by importins-α/β suggests that TPX2 condensation, rather than MT localization, may be regulated. Furthermore, the all-or-nothing nature of inhibition of condensation could explain how TPX2 coordinates a sharp boundary of nucleation with a certain distance from chromatin within the spindle (Fig. 7, above graph)[12,42]. Phase separation as a means to convert gradient signals into binary responses has implications for many biological processes including signaling and morphogenesis[54,55]. Collectively, our findings suggest that TPX2-tubulin co-condensation is a regulated process that may underlie the all-or-nothing activation of branching MT nucleation.

It has been proposed that phase-separated condensates act as reaction crucibles to enhance rates and efficiency[8,9,56]. This conclusion is based on many in vitro studies[3–7,57], but remains

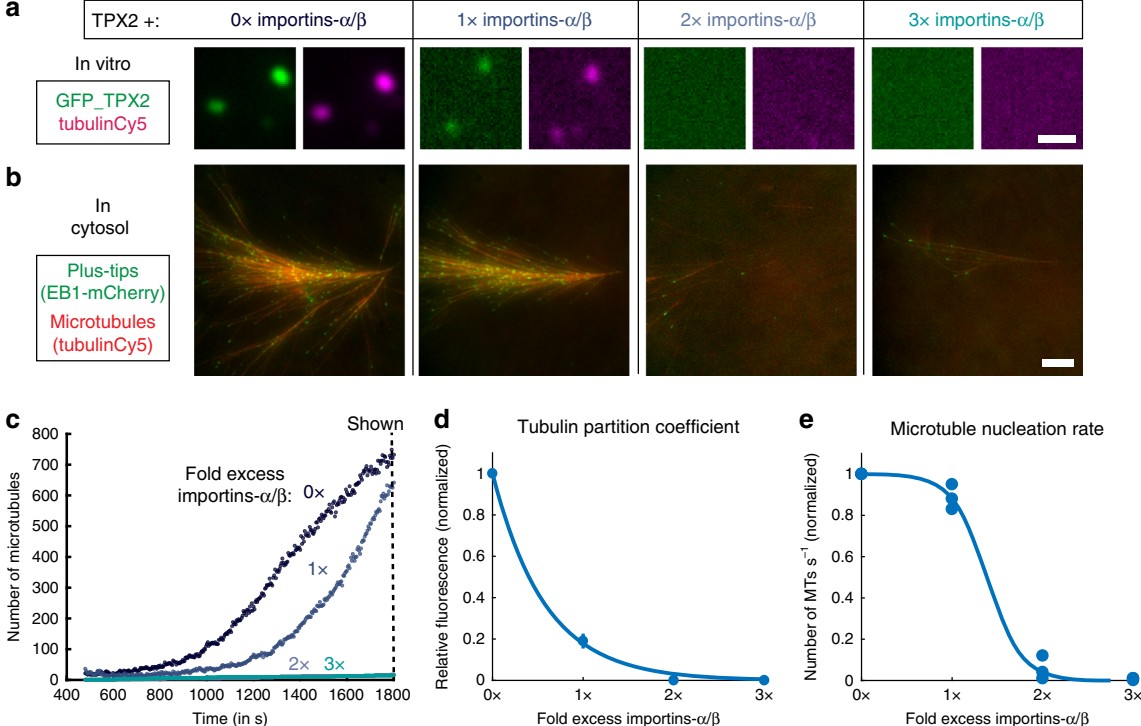

**Fig. 6 Importins α/β inhibit TPX2 condensation and activity. a** Epifluorescent images of TPX2-tubulin co-condensates (green and magenta, respectively) in vitro prepared with importins-α/β at indicated excess (0 × = no importins-α/β) TPX2 and tubulin both at 500 nM. Scale bar, 1 μm. **b** TIRF Images of TPX2-mediated MT nucleation in *Xenopus* meiotic cytosol with TPX2 and importins-α/β added at 100 nM TPX2 and indicated excess of importins-α/β. Cy5-labeled tubulin (red) and mCherry-EB1 (green) highlight microtubules and growing microtubule plus ends, respectively. Images taken at 1800 s. Scale bar, 10 μm. See Supplementary Movie 9. **c** Quantification of data in **b**. **d** Quantification of relative tubulin signal (partition coefficient) as a function of excess importins-α/β. Mean values (points) with ± 1 SD as error bars are shown, *n* = 522 and 420 condensates for 0× and 1× importins-α/β, respectively. **e** Rate of MT nucleation as a function of excess importins-α/β, normalized to 0× importins-α/β. Data pooled from three experimental replicates of (**b**, **c**). Line of best fit shown. Source data are provided as a Source Data file.

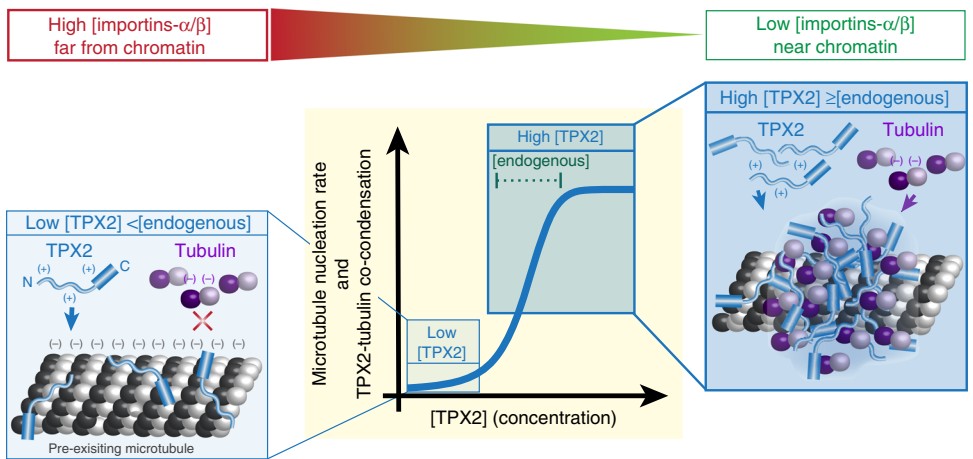

**Fig. 7 Model.** Left-side cartoon: at low concentrations (<endogenous), TPX2 localizes to MTs but does not recruit soluble tubulin, likely due to electrostatic repulsion (denoted by "+" and "−"). Right-side cartoon: at high concentrations of TPX2 (≥endogenous), TPX2 colocalizes with soluble tubulin on microtubules. Center: graphical abstraction of data demonstrating that TPX2 promotes branching MT nucleation (in the cytosol) and forms a co-condensate with tubulin (in vitro) in a switch-like manner at or above its endogenous concentration. Top gradient: relative importin-α/β levels existing as a gradient around chromosomes also affect TPX2-tubulin co-condensation and TPX2-mediated branching MT nucleation.

poorly understood in a physiological context[45,58–60]. We observe a correlation between TPX2-tubulin co-condensation and branching MT nucleation efficiency in the physiological context of *Xenopus* cytosol (Figs. 4 and 5). These results suggest that phase separation of TPX2 and tubulin could underlie the tenfold improvement in the branching MT nucleation efficiency (Fig. 4).

TPX2 selectively enables the autocatalytic amplification of MTs via branching MT nucleation, but does not generate MTs de novo[19,22]. Our data suggests that the specificity of TPX2 to promote branching MT nucleation could result from preferential phase separation onto a pre-existing MT. Lastly, TPX2-mediated branching MT nucleation increases MT nucleation rates up to

100-fold[22] (Table 1), in an all-or-none manner, at a concentration of TPX2 that matches the phase boundary of TPX2-tubulin co-condensation. In summary, we demonstrated that phase separation of a MT nucleation effector may underlie its ability to enhance both reaction efficiency and rate, while also spatially coordinating activity. Phase separation-mediated mechanisms are suspected to be at play in a broad set of cellular processes[9] and this will be important to explore in physiological contexts.

## Methods

**Contact for reagent and resource sharing**. Further information and requests for resources should be directed to and will be fulfilled by lead contact Dr. Sabine Petry (spetry@princeton.edu).

**Protein constructs, expression, and purification**. DH5α cells (New England Biolabs (NEB): C2987I) were used for all subcloning steps. Rosseta2 cells (Fisher: 71-403-4) were used to express proteins for purification. Cells were grown at various volumes in LB Broth (Sigma: L3522) prepared according to the supplier's instructions.

The bovine versions of bovine serum albumin (BSA) (Fisher: 23209) and Tubulin (PurSolutions LLC: 032005) were acquired directly from vendors. Besides the human proteins for FUS (NCBI Gene ID: 2521) and Ran (NCBI Gene ID: 5901), all remaining proteins are X. laevis versions: TPX2 (NCBI Gene ID: 398174), Importin-β (NCBI Gene ID: 100137718), Importin-α (NCBI Gene ID: L36340.1), BuGZ (NCBI Gene ID: 380016), and TOG 1 and 2 from XMAP215 (NCBI Gene ID: 779014). DNA sequences were sourced from in-house plasmids, the X. laevis Gene Collection (Source Biosciences), synthesized (GenScript, Sigma), or were gifted (FUS-IDR plasmid was a gift from Cliff Brangwynne).

All TXP2 constructs were cloned as N-terminally tagged *Strep6xHisGFP-TEV-TPX2* fusions using a modified pST50 vector[61] and were cloned via Gibson assembly (NEB: E2611L). An identical strategy was used to generate *StrHisTEV-mCherry-FL_TPX2*, *StrHisBFP-TEV-IDR_TB*, *StrHisBFP-TEV-RanQ69L*, *EB1-mCherry6xHis* (27), *GST-importin α*, and *GST-importin β*. Insert fragments were PCR amplified from plasmids containing the indicated gene, which was unmodified from its wild-type sequence. Exceptions are as follows: Ran, which has a single site mutation Q69L to render it dominant positive, and *NTΔYF_TPX2* and *Syn_Pos-CT_TPX2*, which were both custom synthesized (NTΔYF) and Sigma (Syn_Pos) custom order. The Syn_Pos (synthetic positive) protein sequence is AKKRKAGDSEGSEGAKKRKAAKKRKAGDSEGSEGAKKRKA. This sequence was generated by interspacing nuclear localization-like sequences (e.g., KKRK) with inert a.a.'s G, S, A, as well as, oppositely charged E. The construct was designed to have a similar theoretical isoelectric point (pI) to the endogenous NT1-480aa of TPX2. All constructs were fully sequenced and were confirmed to have no errors.

All constructs were transformed into Rosseta2 *Escherichia coli* cells and were grown in temperature-controlled incubators shaking at 200 r.p.m. For protein expression, cells were grown at 37 °C (0.5–0.7 $OD_{600}$), then cooled to 16 °C and induced with 0.75 mM isopropyl-β-D-thiogalactoside for another 7 h at 27 °C. Cell pellets were collected and flash frozen for future protein purification.

For all TPX2 constructs, cells were lysed using an EmulsiFlex (Avestin) in lysis buffer (0.05 M Tris-HCl, 0.015 M Imidazole, 0.75 M NaCl, pH 7.75) containing 0.0002 M phenylmethylsulfonyl fluoride (PMSF), 0.006 M β-mercaptoethanol (βME), cOmplete™ EDTA-free Protease Inhibitor tablet (Sigma 5056489001), and 1000 U DNase I (Sigma 04716728001). Lysate was centrifuged in a F21-8 × 50 y rotor using a Sorvall RC6+ at 38,700 × *g* for 25 min. Clarified lysate was bound to pre-washed Ni-NTA agarose beads (Qiagen 1018236) in a gravity column and beads were washed with 10 column volumes lysis buffer. Protein was eluted in lysis buffer containing 200 mM Imidazole and then further purified via gel filtration (Superdex 200 HiLoad 16/600, GE Healthcare—28-9893-35) in CSF-XB buffer (0.01 M Hepes, 0.002 M MgCl, 0.0001 M CaCl, 0.004 M Ethylene glycol-bis(2-aminoethylether)-N,N,N′,N′-tetraacetic acid (EGTA), 10% w/v sucrose, pH 7.75) containing either 0.1 M KCl for extract assays or 0.5 M KCl for condensate assays. Peak fractions were pooled, concentrated (Amicon Ultra, ThermoFisher—various sizes), flash frozen, and stored at −80 °C. Untagged TPX2 was generated by cleaving Strep6xHisGFP-Tev-FL protein with TEV protease at 100:1 TPX2:TEV protease molar ratio overnight at 4 °C, while dialyzing into cleavage buffer (0.02 M NaPO₄, 0.5 M NaCl, 0.006 M βME, and 0.0002 M PMSF, pH 7.5). Cleaved TPX2 was collected as the flow-through of the reaction mixture added to Ni-NTA agarose beads.

For purification of GST-importin α and GST-importin β, clarified lysates were prepared in the same way with the exception of the lysis buffer (0.05 M Tris-HCl, 0.138 M NaCl, 0.0027 M KCl pH 8) and they were bound to a GST affinity column (GSTrap™ Fast Flow, GE Healthcare: 17-5131-02). The column was washed (0.02 M NaPO₄, 0.15 M NaCl, 0.006 M βME, and 0.0002 M PMSF, pH 7.5), protein was eluted (0.02 M Tris-HCl, 0.15 M NaCl, 0.01 M L-Glutathione, 0.006 M βME, and 0.0002 M PMSF, pH 7.5) and peak fractions were pooled. GST-importin β was cleaved to remove GST and was purified in the same manner as untagged TPX2. Strep6xHisBFP-RanQ69 and EB1-6xHismCherry were purified as described previously[19,23]. Pure tubulin and BSA were labeled with commercial NHS-

conjugated dyes (Cy5 or Alexa568) according to the supplier's instructions (Sigma: GEPA150101). A similar method was used to conjugate biotin to tubulin (EZ-Link™ NHS-PEG4-Biotin, ThermoFisher: 21329). Conjugated tubulin was further purified for MT competent tubulin by a series of MT polymerization and pelleting rounds. The percentage of labeling was ≥60% for all purifications. In direct comparisons of fluorescently conjugated proteins (i.e., BSACy5 and tubulin-Cy5), batches were used that had matched the percentage of labeling.

All proteins were flash frozen in CSF-XB buffer at either 0.1 M of 0.5 M KCl and stored at −80 °C. Before use, all proteins were pre-cleared of aggregates via ultracentrifugation at 346,000 × *g* for 15 min in a TLA100 rotor in an Optima MAX-XP ultracentrifuge at 4 °C. Protein concentrations were determined by Coomassie blue densitometry measures of a concentration series of the protein of interest and a BSA standard run on the same SDS-polyacrylamide gel electrophoresis (PAGE) gel.

**Image collection and processing**. The imaging technique used is indicated in each corresponding figure legend. Total internal reflection fluorescence (TIRF), epifluorescence, and differential interference contrast microscopy methods were carried out on a Nikon TiE microscope with a ×100, 1.49 numerical aperture (NA) oil-immersion objective and an Andor Zyla sCMOS camera. Confocal microscopy was carried out on a Nikon TiE microscope with a ×63, 1.45 NA oil-immersion objective, a Yokogawa CSU 21 disk module, and a Hamamatsu ImageEM EM-CCD camera. FRAP was carried out on a Nikon A1 point scanning microscope using a ×63, 1.45 NA oil-immersion objective. All experiments using *Xenopus* egg cytosol were carried out at 18 °C in a temperature-controlled room.

NIS-Elements software was used for all image acquisition. All images within a data set were taken with identical imaging parameters. Binning was not used, except in the case of extract branching MT nucleation assays. FIJI and MATLAB were used for all image analysis. Images within a figure panel were processed with matched brightness and contrast windows for each color to allow direct comparison of intensities. In a few cases, enhanced contrast was used to emphasize non-association events, in which case it was indicated in the legend. These cases include Fig. 2a and Supplementary Figs. 1d and 3d, f, which have enhanced contrast as the main image only (Fig. 2a and Supplementary Fig. 3f) or have an enhanced contrast image in addition to the matched contrast image (Supplementary Figs. 1d and 3). All images are representative crops. Subtraction of background signal was not used, except in the case of GFP_TPX2 signal in Fig. 1i. Oblique-TIRF was used in these and other cases (indicated in figure legends) to visualize GFP-TPX2 signal, which cannot be seen in TIRF due to high levels of TPX2 bound to the coverslip.

**Condensate (phase-separation) assays**. For the standard phase-separation assay, proteins were diluted to 5× the final concentration in a CSF-XB Buffer containing 500 mM KCl salt, then diluted 1:4 in CSF-XB containing no salt at 4 °C to reach a final salt concentration of 100 mM. The reaction mixture was immediately pipetted into a 5 μL volume coverslip flow chamber (constructed with double-stick tape, a glass slide, and a 22 × 22 mm coverslip). The slide was placed coverslip-side down into a humidity chamber for 20 min at room temperature, to allow condensates to settle; reaction was then imaged. Multiple mounting strategies were explored, all yielding similar results. For condensates containing more than one protein, the molar ratio was always 1:1, unless otherwise indicated. Crowding agents were never used.

Partition coefficient is defined as the difference in mean intensity of a condensate compared with background, i.e., apparent relative enrichment. The "Color Threshold" (Otsu thresholding) and "Particle Analyzer" functions on FIJI were used to identify and quantify condensate intensity, respectively. For each concentration, construct, and condition, at least 100 condensates were analyzed. All statistical analysis and graph generation were carried out in MATLAB.

For live observation of condensate formation, fall down, and fusion, coverslip-bottomed CultureWell (Grace BioLabs: 112359) containing CSF-XB buffer with no salt was positioned and focused on a microscope and acquisition was started. A 10× concentration of protein was diluted directly into the well, to achieve a 1:9 dilution and a final salt concentration of 100 mM. Time 00:00 (minutes:seconds) corresponds to the addition of the protein.

To measure FRAP, condensates were prepared as in the live fall-down assay and were allowed to settle for 5 min. Focus was set just above the coverslip and three regions of interest (ROIs) of equal size and geometry were placed (1) within the condensate to be photobleached, (2) the background, and (3) within a nearby condensate. Photobleaching was carried out and the intensity of each ROI recorded every second for the first 10 s and every 10 s thereafter over a 10- to 20-min acquisition. Recovery in the photobleached ROI was normalized to any changes in intensity in the background and nearby condensate ROI due to global bleaching. Global photobleaching in excess of 5% of the starting intensity was never observed. If the intensity in the nearby condensate ROI changed, that FRAP acquisition was discarded.

MT polymerization in vitro was measured as follows. Coverslip flow chamber was first blocked with κ-casein (Sigma: C0406-1G) and then coated with PEG-biotin (Rapp Polymere: 133000-25-20) and NeutrAvidin™ (ThermoFisher: A2666), each in BRB80 buffer (0.08 M Pipes, 0.001 M MgCl₂, 0.001 M EGTA, pH 6.8), and with a BRB80 wash in between each step. Condensates containing tubulin and

TPX2 were prepared as in the standard assay but in this case diluted into BRB80 buffer containing oxygen scavengers and 1 mM GTP for a final salt concentration of 50 mM KCl and 80 mM Pipes. Final concentrations were 10 µM bovine tubulin with 10% labeled Cy5-tubulin, 2% biotinylated tubulin, and 1 µM GFP-TPX2. Condensate mixture was pipetted into the coverslip flow chamber and imaged.

To measure TPX2 soluble pool concentration after phase separation, which corresponds to the phase boundary, condensates were prepared as in the standard assay at concentrations indicated. Samples were then spun at $346,000 \times g$ for 15 min in a TLA100 rotor in an Optima MAX-XP Ultracentrifuge and gel samples were prepared from supernatants (soluble pool). Soluble pool sample concentrations were obtained via densitometry measurements of silver-stained SDS-PAGE.

To measure the phase boundary of TPX2 as a function of [salt] and its concentration, condensates were prepared as in the standard assay at the protein and salt concentrations indicated. A particular [salt]:[TPX2] condition was scored to have phase-separated condensates if the average signal in apparent condensates was at least four times higher than background (i.e., partition coefficient ≥ 4). Conditions were scored to not contain condensates if they fell below this cutoff (only occurred in a 3 of 20 [salt]:[TPX2] conditions) or if no apparent condensates were observed (majority of cases).

**Microtubule localization assays**. To obtain a concentration series of [TPX2] localized to stabilized MTs, 50 µL of a dilute solution of GMPCPP-stabilized MTs containing 10% Alexa568-labeled tubulin were mixed with a 50 µL solution of non-phase-separated TPX2 (GFP-tagged) and tubulin (Cy5 Labled), both at equal molar concentrations in 0.1 M KCl CSF-XB buffer containing 2 mM GTP. Mixture was immediately spun onto Poly-L-Lysine (Sigma: P8920)-coated round coverslips through a 20% glycerol cushion at $25,400 \times g$ in an HB-6 swinging bucket rotor using a Sorvall RC6 + centrifuge. Coverslips were mounted with ProLong® Diamond Antifade Mountant (ThermoFisher: P36970), which was allowed to cure before imaging. To visualize colocalization of TPX2 and soluble tubulin to MTs, samples were prepared in the same way and protein mixtures contained equal molar concentrations of non-phase-separated proteins (TPX2, tubulin, BSA, and/or GFP, as indicated in figure legend).

TPX2 localization to MTs was monitored over time by first attaching 60 µL of a dilute solution of GMPCPP-stabilized MTs containing 10% Alexa568-labeled tubulin and 10% biotinylated tubulin to the surface of a blocked (κ-casein) coverslip-bottomed CultureWell (Grace BioLabs: 112359) via anti-biotin antibodies (Invitrogen 03-3700; 1:10 dilution of stock at 1 µg/µL). The microscope stage was positioned and focused on MTs, after which acquisition started. Fifteen microliters of a 5× concentration of non-phase-separated TPX2 was diluted directly into the well containing 0 M KCl CSF-XB buffer with 2 mM GTP, 1% glucose, and oxygen scavengers to achieve 1× TPX2 concentration and a final salt concentration of 0.1 M KCl. Time 00:00 corresponds to the frame at which TPX2 was added.

To obtain MT depolymerization, 60 µL of a dilute solution of Taxol-stabilized MTs containing 10% Alexa568-labeled tubulin and 10% biotinylated tubulin were attached via anti-biotin antibodies adhered to the surface of a blocked (κ-casein) coverslip-bottomed CultureWell (Grace BioLabs: 112359). The microscope stage was positioned and focused on MTs and the buffer in the well was exchanged three times to remove soluble taxol. A final buffer (60 µL) of 0 M KCl CSF-XB buffer with 2 mM GTP, 1% glucose, and oxygen scavengers was added to the well. Fifteen microliters of a 5× concentration of non-phase-separated TPX2 (or no protein) was added to a final salt concentration of 0.1 M KCl. Initial time (00:00) corresponds to when samples were first imaged after final buffer was added.

FRAP of TPX2 and tubulin localized to MTs was carried out in the same way as for monitoring TPX2 localization over time. TPX2 and soluble tubulin-coated MTs were prepared in the same way as for monitoring TPX2 localization over time. TPX2 and tubulin were allowed to accumulate on MTs for 5 min before photobleaching. Time 00:00 corresponds to the initial photobleaching time point. MTs that were fully, evenly coated with TPX2 and tubulin were selected for photobleaching, which was achieved with one to three well spaced single pixel line(s) perpendicular to the MT. Owing to the different properties of Cy5-tubulin or GFP-TPX2 (organic dye vs. fluorescent protein and different wavelengths relative to the photobleaching laser (405 nm)), FRAP measurements required distinct photobleaching conditions. A 405 nm, 100 mW laser was used in both instances. For bleaching GFP-TPX2, 1 µs dwell time and 15% laser power were used; for bleaching Tubulin-Cy5, 10 µs dwell time and 60% laser power were used. Five images were recorded before photobleaching and ~200–400 images were taken after, each at 1 s intervals. Average intensities of the photobleached field, a non-bleached MT, and the background were recorded over time, post-acquisition, using FIJI. The reported fluorescence recovery was normalized against the background level of continual accumulation of TPX2 and tubulin (derived from the soluble pool) on a nearby non-bleached MT.

**Xenopus egg cytosol assays**. Cytosol naturally arrested in meiosis II was prepared as described in ref. [62]. Stage VI (mature, meiotically arrested) X. laevis eggs were collected after an overnight laying period. Eggs from individual frogs were kept separate but prepared in parallel, and typically two to three batches of eggs were used. Egg jelly coats were removed and the cytosol was fractionated away from the egg yolk, membranes, and organelles by centrifugation ($18,400 \times g$ in HB-6 for 15 min). Eggs were constantly maintained at 18 °C via preparation in a temperature-controlled room. Undiluted cytosol was collected, supplemented with Cytochalasin-D, protease inhibitors, ATP, and creatine phosphate, and was kept at 4 °C until use. All reactions shown within a single image panel were acquired from the same cytosol preparation imaged in a single session.

Preparation of Xenopus cytosol immunodepleted of TPX2 was carried out as described in ref. [58]. Immunoaffinity-purified antibodies against TPX2 (Custom antibody) or an unspecific IgG (Sigma I5006) control antibodies were conjugated to magnetic Dynabeads Protein A (ThermoFisher: 1002D) at 4 °C overnight. Thirty-six micrograms of each antibody was used, each diluted to 0.2 µg/µL. Antibody-conjugated beads were split into two equal volume aliquots; supernatant was removed from one aliquot using a magnetic block and Xenopus cytosol was added. Beads were gently suspended in cytosol every 10 min for 40 min. Cytosol was removed from beads (using magnetic block) and then subjected to another round of depletion using the same procedure with the second aliquot of antibody-conjugated beads. Immunodepletion was assessed via functional assays.

Branching MT nucleation reactions were carried out as described in ref. [63]. Xenopus cytosol was supplemented with fluorescently labeled tubulin ([1 µM] final) to visualize MTs, mCherry-fused EB1 ([100 nM] final) to track MT plus ends, and sodium orthovanadate ([0.5 µM] final) to inhibit motors and prevent MT gliding. Non-phase-separated purified TPX2 constructs (and other proteins) were added at specified concentrations. In the case of +nocodazole, it was added at a final concentration of 0.3 mM. In experiments using immunodepleted cytosol, a constitutively active form of RanGTP (Ran^Q69L) ([7.5 µM] final) was added to prevent sequestration of TPX2 by endogenous importins. In all experiments, 0.1 M KCl CSF-XB buffer was used to match total dilution (25% of extract volume across all experiments). The reaction mixture was prepared on ice, then pipetted into a coverslip flow chamber at 18 °C, which marked the start of the reaction. Reaction was imaged via TIRF microscopy for 20–40 min. All reagents used were in 0.1 M KCl CSF-XB buffer.

In all cases, at least three replicates of the concentration series for each construct was carried out. In these replicates, concentrations were tested in parallel on a single slide setup with multiple flow chambers and imaged at time intervals of usually 30 s to 5 min intervals over the course of 30 min. These intervals are longer than the 2 or 4 s time intervals that are shown in the main figures. In these replicate experiments, multiple constructs or concentrations could be assessed in parallel using a single cytosol prep, which has a finite lifetime (~several hours). These replicates served to verify the patterns observed in the time-lapse Supplementary Movies required for MT nucleation rate analysis described below.

A custom MATLAB software[19] was used to measure the number of MTs over time. EB1 signal on MT plus ends in the entire field of view was used to determine MT number within a single frame. EB1 detection was achieved via the plus-end tracking module of uTrack[64] applied to the entire Supplementary Movie (typically 400–750 frames, 2 or 4 s/frame). Parameters were optimized for each Supplementary Movie according to visual assessment of tracking accuracy. MT nucleation curves were generated by plotting the number of EB1 detections per frame over time. Branching MT nucleation rate is defined as the slope of the linear portion of the MT nucleation curve—i.e., the initial lag and eventually saturation are not used. In instances where branching MT nucleation was not observed (i.e., constant rate of de novo nucleation), rate corresponds to the slope of the entire MT nucleation curve. This quantification method is consistent with previous publications[19,20,23].

For each construct, a number of concentrations were quantified to generate MT nucleation rates within that series. Rates were normalized to the maximum rate within the series and plotted as a function of concentration. A logistic regression equation and the curve fitting tool in MATLAB were used to derive line of best fit. In these experiments, time-lapse Supplementary Movies (~30 min in duration) are acquired at 1 frame/2 or 4 s for each concentration, for each construct. Typically, two to six Supplementary Movies can be acquired per cytosol, given its lifetime. These Supplementary Movies are used to carry out MT nucleation rate analysis described above. For most constructs, at least two independent concentration series and MT nucleation rate quantifications were carried out. Exceptions are IDR_NoTB-CT-TPX2 and TB_IDR-CT-TPX2, where a single concentration series (using time-lapse Supplementary Movies) and quantification was carried out. It is noteworthy that all quantification of every construct was verified with at least three independent replicates (independent cytosol preparations), as mentioned above.

For experiments involving the cytosol overlaid onto condensates, 1 µL of TPX2 condensates, prepared as described in the standard assay, were pipetted onto the center of an untreated coverslip and overlaid with 5 µL Xenopus cytosol (containing mono-dispersed/non-phase-separated Alexa568-labeled tubulin). Extract overlay was carried out either immediately or after condensates were aged for the indicated amount of time (Supplementary Fig. 2d–f). A slide was gently placed on top of the mixture, which marked the start of the reaction (time 0 s). The sample was imaged via oblique-TIRF microscopy. Condensate and branched MT network mass in Supplementary Fig. 2b were calculated by creating fixed size convex hulls around each and conducting integrated intensity measurement across all frames.

**Computational analyses and data visualization**. Disorder prediction was carried out in IUPred[65], which generates a per amino acid prediction of disorder on a scale of 0 to 1. This data was exported and converted into a heatmap using the heatmap function on MATLAB. Secondary structure predictions of TPX2 were carried out

previously[23] using the JPred online tool, specifically the Jpred4 version, which uses the JNet v.2.3.1 prediction program[66]. These predictions, combined with previous functional characterizations of TPX2 domains[23], were used to guide the construction of relevant truncation, mutation, and chimera constructs.

All plots (graphs) were generated in MATLAB. All graphics were made in Adobe illustrator.

**Quantification and statistical analysis**. All number or replicates (*n*) and statistical analyses are indicated in corresponding figure legends.

*In vitro assays*. For all in vitro assays, at least three independent replicate experiments were conducted. Either all quantifications are pooled and displayed, or quantifications from a single representative experiment are shown, indicated in the figure legends. In the latter cases (e.g., partition coefficient measurements), data presented corresponds to the images shown (if applicable).

*Xenopus egg cytosol assays*. All experiments using *Xenopus* egg cytosol were reproduced at least three times using separate cytosol preparations. Similar results were seen in all replicates.

**Experimental organisms used**. DH5α cells were used for all subcloning steps. Rosseta2 cells were used to express proteins for purification. Cells were grown at various volumes in LB Broth (Sigma: L3522) prepared according to the supplier's instructions.

Mature (3- to 7-year-old) female *X. laevis* frogs laid eggs that were used for cytosolic extract experiments. Animal housing, maintenance, and egg harvesting were all carried out to highest ethical standards, in accordance with and approved by the Institutional Animal Care and Use Committee (IACUC) at Princeton University. Petry lab protocol number 1941; IACUC contact information https://ria.princeton.edu/animal-care-and-use.

## Code availability

The code for the custom MATLAB microtubule tracking software is published in and available through Thawani et al.[22]

## Data availability

All data used to support the findings of this study are available, upon request, from the corresponding author. The source data underlying the following figures are provided as a Source Data file: Figs. 1d, e, i; 2e, g, 3b, c; 4c, d; 5b, c; 6c–e; Supplementary Figs. 1c; 3b, c; 4b; 5a–f; 6a–g, 7c–f.

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

## Acknowledgements

We thank members of the Petry Lab for helping with this work, including Ray Alfaro-Aco, Michael Rale, Mohammad Safari, Akanksha Thawani, and Sagar Setru. We are especially grateful to Cliff Brangwynne for experimental suggestions and the Petry Lab, Ibrahim Cisse, and Kassandra Ori-McKenney for feedback on the manuscript. This work was supported by a Ph.D. training grant T32GM007388 by NIGMS of the National Institutes of Health (to M.R.K.), as well as the New Innovator Award of NIGMS of the National Institutes of Health (DP2), the Pew Scholars Program in the Biomedical Sciences, and the David and Lucile Packard Foundation (all to S.P.).

## Author contributions

M.R.K. and S.P. conceived the project, designed the experiments, and wrote the manuscript. M.R.K. generated and characterized reagents and tools, and performed and analyzed experiments.

## Competing interests

The authors declare no competing interests.
