## [Peer Review File · Nature Communications]

Reviewers' comments:

Reviewer #1 (Remarks to the Author):

In the paper "Phase separation of TPX2 enhances and spatially coordinates microtubule nucleation", King and Petry examine the "co-condensation" of TPX2 and tubulin. The authors find that (1) TPX2 co-condensates with tubulin outside of cells, (2) TPX2-tubulin co-condensation preferentially occurs on pre-existing microtubules outside of cells, (3) this co-condensation enhances the efficiency of TPX2-mediated microtubule nucleation, and (4) importin a-b regulates this co-condensation.

While the data in the paper is interesting, it is not clear to me that the authors present a significant advance from previously published work. Specifically, "co-condensation" seems to be very similar to the widely reported "clustering" of TPX2 and tubulin, and, other than developing new terminology for this observation, it is not clear that new molecular mechanisms for TPX2 function are described here.

Specific Comments:

1) The authors note that, when GFP-tagged TPX2 in high salt buffer was introduced to "physiological salt levels", spherical condensates form. However, neither the salt concentrations nor the TPX2 concentrations are listed in the text. The specific conditions used to generate this observation should be included in the main text or figure legend.

2) Further, while the final salt concentration may be physiological, it is not clear that large, spherical condensates of TPX2 could be formed at physiological TPX2 concentrations, or whether the transfer of TPX2 from one salt concentration to another was necessary for this behavior to occur. If this behavior is an artifact of switching from one salt concentration to another at a high TPX2 concentration, it is not clear that this observation is relevant to the cellular function of TPX2.

3) Similarly, the concentration of TPX2 in Fig. 1C (4 μ M), seems to be orders of magnitude larger than physiological concentrations (\sim 0.04 μ M), strongly suggesting that the observations shown here are an artifact of using extremely high TPX2 concentrations.

4) The authors argue that mono-dispersed GTP-TPX2 stimulates the formation of branching MT nucleation by forming co-condensates with tubulin (Fig. 1G,H), however, it is not clear that co-condensates of the size shown in Figs 1D,F are present in this assay. Rather, it is likely that small clusters or aggregates of tubulin and TPX2 have formed, as previously described in multiple papers.

4) The authors quantify TPX2-tubulin co-condensation using more physiologically relevant concentrations of TPX2 in Fig. 2. Here, the authors demonstrate that TPX2 binds to pre-formed microtubules at concentrations as low as 1 nM, however, free tubulin does not bind to the pre-formed microtubule lattice until the concentration of TPX2 and tubulin are ~ 50 nM, which is equivalent to the concentration of TPX2/tubulin that represents the “phase boundary” of TPX2-tubulin co-condensation in solution. Thus, the authors suggest that TPX2-tubulin co-condensates are forming on the microtubules, although they are clearly not of the size as was shown in Fig. 1. Regardless, because TPX2 has two tubulin binding sites, and tubulin subunits can self-associate, it is not clear that “co-condensates” are distinguishable from the clusters or aggregates of TPX2 and tubulin that have already been widely reported, and are related to the relative affinities of the molecules. Thus, I am not convinced that anything new beyond a new descriptive terminology is reported here.

5) The authors state that branching microtubule nucleation is “critical for rapid and accurate spindle assembly”. However, many/most organisms appear to efficiently build spindles in the absence of branching microtubule nucleation. Until this mode of microtubule nucleation is demonstrated as a critical component of spindle assembly across a range of organisms, these types of general statements are inappropriate and very much overstated. All general statements regarding the role of branching nucleation in cellular function should reflect the current state of understanding regarding the role of branching nucleation.

Reviewer #2 (Remarks to the Author):

King and Petry investigate the molecular mechanism of TPX2-dependent microtubule branching in vitro and in xenopus extracts and propose that co-phase separation of TPX-2 and tubulin on the surface of pre-existing microtubules promote branching. They further propose that a known inhibitor of TPX-2, Importin-alpha/beta, inhibits TPX-2-dependent microtubule assembly by suppressing co-phase separation of TPX-2 and tubulin.

Earlier studies have proposed that phase separation can drive microtubule nucleation via concentrating tubulin. This study proposes TPX-2 uses a similar mechanism to generate branched microtubules. The findings of this study will be of significant interest to investigators studying TPX-2, microtubule and spindle assembly and therefore worthy of publication here after the following concerns have been addressed.

Major concerns

1) While the authors clearly demonstrate TPX-2 and tubulin can co-phase separate in vitro and support microtubule nucleation, evidence that the assembly of TPX-2 and tubulin on the surface of pre-existing microtubules is indeed a phase separation process is lacking. Being the central claim of the paper, this must be demonstrated. The authors could consider the following suggestions.

A) Time-lapse imaging to show that co-condensates of TPX-2/tubulin wet and spread on the surface of pre-existing microtubules, when co-condensates and microtubules come in contact with each other.

B) Time-lapse imaging to show that TPX-2/tubulin co-condensates emerge when microtubule coated with TPX-2/tubulin is treated with nocodazole.

C) FRAP experiments to show TPX-2 and tubulin recruited on the surface of microtubule is dynamic.

2) From the elegant structure-function analysis in Figures 2-3, it is clear that phase separation enhances microtubule branching but is not required for a basal level of branching. It is also unclear how the precise branch-points are chosen on the surface of pre-existing microtubules. In the context of the proposed liquid-like TPX-2/tubulin phase, can one expect all points on the surface of a microtubule to have equal chance of becoming a branch point? It would be interesting to look at the distribution of gamma-tubulin ring complex, XMAP215 and augmin in microtubule branching assays in the cytosol.

Minor concerns

1) Consider replacing the term 'mono-disperse' with 'non-phase-separated'.

2) Figure 1A: Difficult to distinguish between low disorder and predicted alpha-helices.

3) Figure 1B: Consider a better DIC image.

4) Define 'partition co-efficient' in the figure itself.

5) Figure 3D and 4C: Why are the 'full-length' data so different? Have these been accidentally misrepresented?

6) The mechanism of how importin alpha/beta inhibits phase separation of TPX-2/tubulin should be discussed – is this a competitive binding mechanism? Are there other examples of such mechanism in phase separation regulation?

7) The abstract should be revised to remove claims not supported by current data.

In the paper “Phase separation of TPX2 enhances and spatially coordinates microtubule nucleation”, King and Petry examine the “co-condensation” of TPX2 and tubulin. The authors find that (1) TPX2 co-condensates with tubulin outside of cells, (2) TPX2-tubulin co-condensation preferentially occurs on pre-existing microtubules outside of cells, (3) this co-condensation enhances the efficiency of TPX2-mediated microtubule nucleation, and (4) importin a-b regulates this co-condensation.

While the data in the paper is interesting, it is not clear to me that the authors present a significant advance from previously published work. Specifically, “co-condensation” seems to be very similar to the widely reported “clustering” of TPX2 and tubulin, and, other than developing new terminology for this observation, it is not clear that new molecular mechanisms for TPX2 function are described here.

We thank the reviewer for their nice summary and for bringing up this concern. We agree with the reviewer that previous reports of TPX2-tubulin ‘clustering’ are similar to our demonstrations of TPX2 tubulin co-condensation in Figure 1. In fact, the whole purpose of Figure 1 is to make it very clear that the previously observed TPX2-tubulin ‘clusters’, which could not be explained beyond their observation, are indeed proper TPX2-tubulin co-condensates. We demonstrate this based on state-of-the art methods accepted in the new field encompassing liquid-liquid phase separations (LLPS).

Laying out that TPX2 and tubulin form co-condensates in Figure 1 was essential to synthesize the series of hypotheses, experiments, and results that provide novel molecular insight into how TPX2 stimulates branching microtubule nucleation (Figures 2-6). Besides explaining how TPX2 contributes via LLPS to branching microtubule nucleation, and thereby spindle assembly, the findings are very important for the field encompassing LLPS. Notably, this is a sought-after demonstration of the hypothesis that LLPS of a single pair of effector (TPX2) and substrate (tubulin) directly affects reaction kinetics in a single reaction within a physiological environment.

In summary, our characterization of TPX2 ‘clusters’ as TPX2-tubulin co-condensates is significant in itself, and was an essential pre-requisite to elucidate a molecular mechanism of how TPX2 stimulates branching microtubule nucleation and spindle assembly.

Specific Comments:

1) The authors note that, when GFP-tagged TPX2 in high salt buffer was introduced to “physiological salt levels”, spherical condensates form. However, neither the salt concentrations nor the TPX2 concentrations are listed in the text. The specific conditions used to generate this observation should be included in the main text or figure legend.

We thank the reviewer for noticing this oversight. We incorporated the specific conditions used to generate TPX2 condensates in the text of the results section, and

clarified how and why the assay was conducted. In addition, we highlighted the specific conditions as a graphic in Figure 1 itself (Fig. 1C). These modifications make relevant experimental details clear for the reader.

2) Further, while the final salt concentration may be physiological, it is not clear that large, spherical condensates of TPX2 could be formed at physiological TPX2 concentrations, or whether the transfer of TPX2 from one salt concentration to another was necessary for this behavior to occur. If this behavior is an artifact of switching from one salt concentration to another at a high TPX2 concentration, it is not clear that this observation is relevant to the cellular function of TPX2.

We thank the reviewer for this comment, which highlights that we have to better explain why these experiments were conducted this way, in particular to an audience who is new to the rapidly growing field of LLPS. There are several reasons why the experiment was conducted in this particular order and with these conditions (listed below). We have modified the results text (pg. 4 paragraph 1 of results section) to make clear that this is a standard and well-validated assay for testing LLPS:

1. We note that TPX2 cannot be purified in physiological salt levels since it will form a LLPS condensate. Standard protein purification methods have been established for monomeric proteins or defined protein complexes, but are incompatible in many aspects for proteins that undergo LLPS (column clogging and precipitation, among others). Instead, it is necessary to purify LLPS proteins in high salt to keep them monomeric. This is also the condition in which the protein is stored, which is necessary to make any meaningful conclusions in the experiments to follow. This is consistent with best practices for purifying phase separating proteins detailed in Alberti S. et al. 2019 (full citations at end of this document).
2. To characterize the behavior of proteins undergoing LLPS, it is standard to start out with their purification condition containing high salt, in which no LLPS can form. The salt concentration is then dropped to assess the protein's ability for phase separation at different salt concentrations. This is the only way to establish if the protein monomers coalesce into a LLPS condensate (or nearly crash out as an aggregate). It is also the typical way that a phase diagram is established, see Fig. S1B. Lastly, we note that this purification scheme and phase separation assay are standard and well-accepted for disordered, phase separating proteins. For further information please see Alberti S. et al. 2018 and 2019.
3. Indeed 'large' (μm sized) and spherical condensates do not form at lower, physiological protein concentrations. This is consistent with the well-established phenomenon in phase separation that lower protein concentrations result in smaller condensates, i.e. condensate size scales with concentration (for examples please see work by Clifford Brangwynne, Anthony Hyman, and Mike Rosen). Generating large spherical condensates in the initial assays of phase separation (Figures 1 and S1) is necessary to validate TPX2 as LLPS protein. Following this assessment, we use the rest of the manuscript to assess whether the phenomena of LLPS is physiologically relevant (which we conclude it is), and in what form

LLPS of TPX2 occurs (clearly not as a large spherical condensate but as a smaller condensate on the MT from which it acts).

4. “Is lowering salt necessary for phase separation to occur?” We found that this is not the case. Supplemental figure 1B shows a salt-phase diagram demonstrating that TPX2 at high concentrations still phase separates at high salt. Additionally, we show in Figure 2C that the mono-dispersed TPX2 forms a spherical condensate directly in isolated cytosol. This occurs only when its preferred substrate is not present, i.e. when microtubule formation is inhibited via Nocodazole. As soon as microtubules are present, TPX2 will bind to microtubules (Figure 2), which is critical for its function as detailed in Figures 3-7.

In summary, these observations and many characteristics of TPX2 (high disorder, multiple binding partners, independent observations of clustering *in vitro* and in cytosol that have been published over the last ~2 decades) collectively suggest that condensation is not an artifact of TPX2 but rather an intrinsic property of the protein, whose functional importance we address in this manuscript. Again, we thank the reviewer for bringing up this concern. In addition to the response above, throughout the manuscript, in particular in the results section, we have clarified why specific conditions were chosen and we contextualize results for an audience who is new to the phase separation field.

3) Similarly, the concentration of TPX2 in Fig. 1C (4 μM), seems to be orders of magnitude larger than physiological concentrations ($\sim 0.04 \mu\text{M}$), strongly suggesting that the observations shown here are an artifact of using extremely high TPX2 concentrations.

We thank the reviewer for pointing this out. We will first explain why a high concentration of TPX2 was chosen to demonstrate LLPS and then address how we clarified this in the manuscript.

1. We agree that the co-condensates shown in figure 1C are large and at a high TPX2 concentration. Showing condensates at a variety of concentrations, including high ones, is standard for the phase separation field and is initially done to highlight the spherical nature of condensates and to characterize the conditions in which LLPS occurs. This can be thought of as a proof-of-principle demonstration of condensation in a clear assay where the phase separation behavior can be easily observed and there is no doubt that it occurs. We strongly agree with the reviewer that conclusions about protein behavior cannot be drawn from these experiments alone. For that reason, the rest of our work, centered on Figures 2 – 7, addresses the physiological nature and function of TPX2 LLPS at or near its endogenous concentration.
2. The experiments that follow figure 1C (namely Figures 2, 3 and S3) establish that co-condensates indeed form at the physiological concentration of TPX2 (as the reviewer notes in comment 4b). We note that it is unusual in the LLPS literature to demonstrate phase separation at the endogenous concentration of the protein and this is therefore a strength of our paper. Moreover, the phase boundary not only coincides with the endogenous concentration, but also the kinetically effective concentration for branching microtubule nucleation, which further

makes this work stand out in the LLPS field, and supports the biological function of TPX2's LLPS.

3. Lastly, and most importantly, we specified in the text that the in-solution droplets (Figure 1) are not observed in the physiological context, and the interesting turn of our research is that TPX2 prefers to bind and condense onto microtubules instead of associating with itself in solution (Figure 2). Importantly, we have carried out several new experiments (namely FRAP and time-course microscopy) that validate TPX2's LLPS nature on MTs (Figure 2 and S4), as requested by another reviewer. These new data significantly improve our manuscript and highlight even more that the physiological form of TPX2-tubulin co-condensates are on the MT lattice. This behavior of condensation on MTs is the key to explain how TPX2 stimulates branching microtubule nucleation (Figures 3-6).

We clarified each of these three points in the manuscript and modified both the results (pg. 9, paragraph 1) and discussion (pg.'s 20-21) sections to focus on the important part of TPX2-tubulin co-condensation, namely that it seems to occur on the microtubule (not as a large droplets in solution) and that it enhances the kinetic efficiency of branching microtubule nucleation and thereby spindle assembly.

4a) The authors argue that mono-dispersed GTP-TPX2 stimulates the formation of branching MT nucleation by forming co-condensates with tubulin (Fig. 1G,H), however, it is not clear that co-condensates of the size shown in Figs 1D,F are present in this assay. Rather, it is likely that small clusters or aggregates of tubulin and TPX2 have formed, as previously described in multiple papers.

We thank the reviewer for this comment, as it highlights that we need to improve clarity in our manuscript, which we addressed below. We believe this concern stems from a misunderstanding about how the data shown in Figures 1D,F as well as 1G,H are connected, specifically how the 'in-solution' condensates (Fig. 1D,F) relate to condensates that form on MTs ('on-MT' condensates) (formerly Fig. 1G,H, now Figure 2A-B). What we tried to convey is that in-solution condensates tell us about the biophysical nature of TPX2, but do not reflect physiologically observed structures. In contrast, we indeed did not observe them in cytosol, rather TPX2 seems to coat MTs as soon as they are present (now Figure 2A-B). This discrepancy prompted us to investigate how TPX2 interacts with the MT, and this is what turned out to be functionally important (Figures 2 – 7).

First of all, we articulated this apparent discrepancy that the reviewer noted as well, in the text, which serves nicely as a transition to the figures that follow. We moved Fig. 1G,H into separate (new) figure 2 that exclusively addresses how TPX2 interacts with the MT. In addition, we carried out new experiments and modified the text to strengthen our argument that TPX2 and tubulin preferentially forms LLPS co-condensates on MTs (results section pg.'s 8-10) and that this represents the physiologically relevant form of TPX2 condensation (new Figure 2). Finally, these points are also revisited in the discussion (pg. 20 paragraph 2). We hope these modifications distinguish in-solution vs. on-MT TPX2 condensates, and help highlight the physiological nature and function of TPX2-tubulin co-condensates on MTs.

4b) The authors quantify TPX2-tubulin co-condensation using more physiologically relevant concentrations of TPX2 in Fig. 2. Here, the authors demonstrate that TPX2 binds to pre-formed microtubules at concentrations as low as 1 nM, however, free tubulin does not bind to the pre-formed microtubule lattice until the concentration of TPX2 and tubulin are ~ 50 nM, which is equivalent to the concentration of TPX2/tubulin that represents the “phase boundary” of TPX2-tubulin co-condensation in solution. Thus, the authors suggest that TPX2-tubulin co-condensates are forming on the microtubules, although they are clearly not of the size as was shown in Fig. 1.

As explained in detail in remarks 3) and 4a), and therefore only summarized here, we made extensive modifications to the manuscript to clarify the distinction between in-solution LLPS of TPX2 (Figure 1) and on-MT LLPS of TPX2 (Figure 2). The in-solution LLPS is a proof-of-principle demonstration of LLPS, in which images of the clearest examples are shown. The important part is that TPX2 forms a condensate on MTs, which is functionally relevant to stimulate branching MT nucleation. While many LLPS have been identified in solution *in vitro*, it was less clear if that was occurring in the cell. We demonstrated that, in fact, TPX2 forms a LLPS on the MT, which also demonstrated that LLPS can form on a 3D substrate (which has also been shown for BuGZ and tau).

Our new data (Fig 2 H-I and S4) indicate that the material state of TPX2 on the MT is similar to that in solution. However a more detailed understanding of this material state, its maturation process, and its contribution to function still needs to be investigated and this is an exciting future research area. In addition to a similar material state between in-solution and on-MT condensates, we also find that they are, in fact, of similar size - so long as the same concentration is being compared. As noted in responses to comments 3 and 4a, condensate size scales with concentration, so it is not fair to compare 4 μ M TPX2 condensates shown in figure 1 to the 0.1-0.25 μ M condensates on MTs shown in Figure 2. To make a clear comparison between in-solution and on-MT condensates at the same concentration we have added Fig. 2F-G (formally Fig. S3D-E), which directly compares the two. As shown in these images, both in-solution and on-MT condensates are of similar size and partition co-efficient (see 100nM and 250nM examples). These new results provide clear evidence and characterize the TPX2-tubulin co-condensates on MTs as manifestations of LLPS and make clear to the reader how condensate size scales with concentration (pg. 9, paragraph 2)

Regardless, because TPX2 has two tubulin binding sites, and tubulin subunits can self-associate, it is not clear that “co-condensates” are distinguishable from the clusters or aggregates of TPX2 and tubulin that have already been widely reported, and are related to the relative affinities of the molecules. Thus, I am not convinced that anything new beyond a new descriptive terminology is reported here.

We thank the reviewer for highlighting these points. Below, we discuss what was known before and how our study furthers knowledge about how TPX2 contributes to branching MT nucleation. We edited the introduction (pg. 3, paragraph 2) and, in particular, the discussion (pgs. 20-21) section to clarify how this manuscript contributes to our understanding of TPX2 and its function.

1. It has been demonstrated multiple times in gel filtration assays that monomeric TPX2 does not bind dimeric tubulin (Alfaro-Aco, et al., 2017 and Roostalu, et al., 2015). Therefore it remained a mystery how TPX2 could interact with tubulin *in vitro*, e.g. in the observed clusters. Our thorough and quantitative characterization of TPX2-tubulin co-condensation now provides a framework for the nature of this interaction.
2. The “two tubulin binding sites” on TPX2 seem to refer to the two sites (ridge and wedge) recently seen in a cryo-EM structure of TPX2 bound to a polymerized MT (Zhang et al., 2017). We note that these binding sites are located at interfaces that are only present on a polymerized MT lattice (not soluble tubulin which monomeric TPX2 cannot bind). Therefore, we disagree with the statement that there exist two tubulin binding sites, rather they are two MT lattice binding sites. More convincingly, TPX2 truncation constructs without these two binding sites (CT_319-716 and CT_480-716) still co-condensate with tubulin (our paper) and are functional (i.e. promote branching MT nucleation in extract). Besides in our Figure 4, this has been demonstrated in Brunet et al., 2004 and Alfaro-Aco et al., 2017
3. Lastly, phase separated proteins rarely use discrete interaction motifs to undergo co-condensation. The interaction mechanism for co-condensing proteins is an area of active study and emerging consensus is consistent with our findings for TPX2, namely that residues distributed throughout the entire length of the protein, especially disordered regions, contribute to interactions/condensation (Figure 4) (Wang et al. 2018). Furthermore, observation in Alfaro-Aco et al., 2017, that longer TPX2 constructs are more efficient at binding to a microtubule, was a hint that a LLPS mechanism is at play in binding to the microtubule.

In summary, we are not aware of evidence in the literature for an affinity / discrete-binding-site mechanism that explains the TPX2-tubulin interaction. Rather, existing evidence in the literature and our own suggest that TPX2 and tubulin interact as a co-condensate, and that this preferentially occurs on a microtubule. The important part is, that besides identifying this in Figures 1 and 2, we can uncover whether and how this is important for branching MT nucleation (a reaction that was only shown to occur in spindle assembly in 2013). We directly addressed these points in the discussion (pg. 22 paragraph 2) to avoid any confusion, and also clarified in the introduction (pg. 3 paragraph 2) to address what was known before about the role of TPX2.

4a) ... “small clusters or aggregates of tubulin and TPX2...previously described in multiple papers.”

4b) ... “it is not clear that “co-condensates” are distinguishable from the clusters or aggregates of TPX2 and tubulin that have already been widely reported,...”

In all comments above, the issue with the terminology versus a real research advance is addressed. In our replies, we stated how we edited the manuscript thoroughly to make clear what the real research advances are. In addition, we provide an appendix here that summarizes previous literature about TPX2 clusters to make it very clear that no other publication reports or characterizes the nature (LLPS) of these “clusters” as our manuscript does. We believe that our manuscript brings a lot of clarity to the

contradicting literature on TPX2, because it can finally explain all the observations within one model, encompassing LLPS, which, importantly, provides a new mechanism by which TPX2 stimulates branching microtubule nucleation.

5) The authors state that branching microtubule nucleation is “critical for rapid and accurate spindle assembly”. However, many/most organisms appear to efficiently build spindles in the absence of branching microtubule nucleation. Until this mode of microtubule nucleation is demonstrated as a critical component of spindle assembly across a range of organisms, these types of general statements are inappropriate and very much overstated. All general statements regarding the role of branching nucleation in cellular function should reflect the current state of understanding regarding the role of branching nucleation.

We thank the reviewer for this comment. Indeed, we should be more precise about which organisms utilize branching MT nucleation and to what extent, and have edited the introduction of manuscript (pg. 3 paragraph 2) accordingly.

Appendix: Previously published observations on TPX2 clusters

- a. Schatz et al., 2003: “amorphous aggregates of varying size, which only formed in the presence of both TPX2 and tubulin. Microtubules emanated randomly in all directions from these aggregates” Figure 3A and Figure 5 are shown, scale bars at 10 μ m and 500nm respectively.

- b. Brunet et al., 2004: “GFP-TPX2 formed aggregates with tubulin from which microtubules emanated forming asters (Schatz *et al.*, 2003). We found that the N-terminal domain TPX (1-480) also formed aggregates and microtubule asters (Figure 3). TPX2 (241-715) formed smaller aggregates from which short bundled microtubules grew. The C-terminal

domain, TPX2 (319-715), formed aggregates, but none of these aggregates nucleated microtubules” Figure 3A shown scale bar is 10µm.

- c. Roostalu et al., 2015: “biotinylated TPX2 induced massive nucleation of slowly growing microtubules. Hence, in solution, TPX2-induced nuclei can transform into microtubules, a conformational change that is apparently inhibited when nuclei form directly on [surface immobilized TPX2]” Figure 6B shown.

Citations:

Alberti, S., Gladfelter, A., and Mittag, T. (2019). Considerations and Challenges in Studying Liquid-Liquid Phase Separation and Biomolecular Condensates. *Cell* 176, 419–434.

Alberti, S., Saha, S., Woodruff, J.B., Franzmann, T.M., Wang, J., and Hyman, A.A. (2018). A User’s Guide for Phase Separation Assays with Purified Proteins. *Journal of*

Molecular Biology 430, 4806–4820.

Alfaro-Aco, R., Thawani, A., and Petry, S. (2017). Structural analysis of the role of TPX2 in branching microtubule nucleation. *J Cell Biol* 216, 983–997.

Brunet, S., Sardon, T., Zimmerman, T., Wittmann, T., Pepperkok, R., Karsenti, E., and Vernos, I. (2004). Characterization of the TPX2 Domains Involved in Microtubule Nucleation and Spindle Assembly in *Xenopus* Egg Extracts. *Mol. Biol. Cell* 15, 5318–5328.

Roostalu, J., Cade, N.I., and Surrey, T. (2015). Complementary activities of TPX2 and chTOG constitute an efficient importin-regulated microtubule nucleation module. *Nat. Cell Biol.* 17, 1422–1434.

Schatz, C.A., Santarella, R., Hoenger, A., Karsenti, E., Mattaj, I.W., Gruss, O.J., and Carazo-Salas, R.E. (2003). Importin alpha-regulated nucleation of microtubules by TPX2. *EMBO J.* 22, 2060–2070.

Tulu, U.S., Fagerstrom, C., Ferenz, N.P., and Wadsworth, P. (2006). Molecular Requirements for Kinetochores-Associated Microtubule Formation in Mammalian Cells. *Curr Biol* 16, 536–541.

Wang, J., Choi, J.-M., Holehouse, A.S., Lee, H.O., Zhang, X., Jahnel, M., Maharana, S., Lemaitre, R., Pozniakovskiy, A., Drechsel, D., et al. (2018). A Molecular Grammar Governing the Driving Forces for Phase Separation of Prion-like RNA Binding Proteins. *Cell* 174, 688-699.e16.

Zhang, R., Roostalu, J., Surrey, T., and Nogales, E. (2017). Structural insight into TPX2-stimulated microtubule assembly. *ELife Sciences* 6, e30959.

Major concerns

1) While the authors clearly demonstrate TPX-2 and tubulin can co-phase separate in vitro and support microtubule nucleation, evidence that the assembly of TPX-2 and tubulin on the surface of pre-existing microtubules is indeed a phase separation process is lacking. Being the central claim of the paper, this must be demonstrated. The authors could consider the following suggestions.

We thank the reviewer for this insight and these excellent suggestions. Below we detail the new data that we generated to address each of these suggestions. Overall, we demonstrate that TPX2 and tubulin on MTs exhibit several features of phase separation including formation of spherical co-condensates, fusion of co-condensates and fluorescence recovery after photo bleaching. These findings are integrated into figure 2 and a new supplemental figure (Fig. S4). We believe these new findings further the manuscript and its impact, as phase separation on MTs is still a very new concept and these results clarify that it indeed takes place.

A) Time-lapse imaging to show that co-condensates of TPX-2/tubulin wet and spread on the surface of pre-existing microtubules, when co-condensates and microtubules come in contact with each other.

To address this suggestion of time-lapse imaging, we designed the following experimental set-up: a mixture of TPX2 and tubulin are added to a coverslip bottomed chamber with MTs pre-attached and the resulting dynamics and binding of TPX2 and tubulin to the MT are observed live (see new Fig. 2H for a schematization). This experimental set-up is similar to our previously used fixed time-point assay. Both the live and fixed time point assay give similar results: at concentrations above the phase boundary for in-solution TPX2-tubulin phase separation, spherical co-condensate structures form on the surface of MTs (Figure 2). Results from the live assay, furthermore, show that co-condensates fuse on the surface of MTs (Fig. 2I), which is indicative of a liquid-like nature.

A detailed examination of the time course of TPX2-tubulin localization to MTs is shown in fig 2I and Supplemental Movie 4. These data reveal a sequential pattern of events that are consistent with TPX2-tubulin liquid-like phase separation on a MT. Initially TPX2 in the soluble pool rapidly and indiscriminately localizes along the MT lattice. Then, as more TPX2 accumulates, TPX2 beads up into co-condensates along to the MT lattice. Only when TPX2 condensates have formed on the MT, we observe tubulin co-localizing with TPX2 condensates on the MT, further indicating that the two proteins exclusively interact as LLPS co-condensates. Condensates in close proximity on the MT fuse, providing strong evidence for their liquid state. This sequence of events is reminiscent of the well-known phenomenon of Rayleigh instability, which explains why liquids on a cylindrical surface bead up, such as dew on a tree branch (Mead-Hunter, R., *Langmuir*, 2012)

B) Time-lapse imaging to show that TPX-2/tubulin co-condensates emerge when microtubule coated with TPX-2/tubulin is treated with nocodazole.

We agree with the reviewer that depolymerizing a MT and observing the emergence of a spherical TPX2-tubulin co-condensate would be a useful additional demonstration of the liquid-like nature of TPX2-tubulin co-condensation on MTs. We furthermore agree that observing this *in vitro* would allow to visualize a re-condensation process live. In contrast, currently we demonstrate that, in cytosol, Nocodazole treatment leads to the appearance of spherical TPX2-tubulin co-condensates (Fig. 2C). Nocodazole was used in the cytosol assay to prevent MT polymerization, since it was most appropriate pharmacologically drug and had the added benefit of demonstrating that monodispersed TPX2 and tubulin can find each other and coalesce via LLPS in a complex cytosol (Fig. 2C).

Direct addition of Nocodazole to MTs *in vitro* failed to depolymerize already formed MTs in our hands, which is consistent with the ability of Nocodazole to prevent the re-polymerization of dynamic MTs but not to depolymerize stable MTs (Gayek and Ohi, R 2015). Therefore, we employed a Taxol washout strategy in this *in vitro* assay to actively depolymerize MTs (Fig. S4B).

As we show in supplemental figure 4A, we identified a Taxol washout condition in which MTs readily depolymerize. However, TPX2-covered MTs are protected from depolymerization, which was somewhat unexpected, but consistent with previous observations of TPX2 as an “anti-catastrophe” (Reid, T.A., et al., *JCS*, 2016) or “stabilization” (Zhang R., et al., *eLife*, 2019) factor.

Therefore, the nature of TPX2-mediated stabilization of the MT limits the outcome of this experiment, which in the end still provided an interesting result that we reported. Importantly, we nevertheless provide several lines of evidence demonstrating that TPX2 and tubulin form liquid-like co-condensates on MTs.

C) FRAP experiments to show TPX-2 and tubulin recruited on the surface of microtubule is dynamic.

This is also an excellent suggestion and we thank the reviewer for it. Having established a time-lapse set up for monitoring TPX2-tubulin recruitment to the MT (previous excellent suggestion), we were able to conduct FRAP on these samples.

As shown in supplemental figure 4B, we find that TPX2 is indeed dynamic on MTs and shows a FRAP recovery profile similar to TPX2 condensates in solution (Fig. S1C). Importantly, we normalized the recovery of fluorescence intensity against a background level of continual accumulation of TPX2 onto MTs from the soluble pool. This normalization isolates the recovery that is due to TPX2 that is already localized to the MT and whose dynamic, and as we argue liquid-like, nature permits it to diffuse along the MT lattice.

2) From the elegant structure-function analysis in Figures 2-3, it is clear that phase separation enhances microtubule branching but is not required for a basal level of branching. It is also unclear how the precise branch-points are chosen on the surface of

pre-existing microtubules. In the context of the proposed liquid-like TPX-2/tubulin phase, can one expect all points on the surface of a microtubule to have equal chance of becoming a branch point? It would be interesting to look at the distribution of gamma-tubulin ring complex, XMAP215 and augmin in microtubule branching assays in the cytosol.

We thank the reviewer for these insightful queries, which we address individually below.

(a) Can one expect all points on the surface of a MT to have equal chance of becoming a branch point?

Yes. First of all, we do not see any bias on the MT lattice as to where the TPX2-tubulin co-condensates form and in the new manuscript version comment on that in the results and discussion section (see figure 2D-I for examples and Discussion pg. 20, paragraph 2).

Second, after having understood how TPX2 works thanks to this manuscript, we were able to fully reconstitute branching MT nucleation *in vitro* with augmin, the gamma-tubulin ring complex, XMAP215 and TPX2 (Alfaro-Aco et al, *bioRxiv* 2019), something that we have tried to achieve over at least 6 years. Indeed, new MTs grow out of the TPX2 condensates along the MT lattice (see panel D below). Most importantly, all sites on the lattice have an equal probability of serving as a branch point (see panel C below). A reference to this work, which further supports no positional bias, is now added in the Discussion.

Third, the question remains whether there is still an equal chance of becoming a branch point in cytosol. A study from our lab has meanwhile been published (Thawani et al., *eLife* 2019) that addresses this point. Here, we show that branching events are biased toward the minus end (i.e. older end) of the pre-existing MT. This is due to the fact that

the regions toward the minus end are available for a longer amount of time, which increases the amount / probability for TPX2 to bind. In that study the authors also clearly show that this is not due to any other feature of the MT (Thawani et al., *eLife* 2019). A reference to this work is now also included in the Discussion.

(b) What about the distribution of gamma-tubulin ring complex, XMAP215 and augmin in microtubule branching assays in the cytosol?.

This question is a very important one and has been addressed in separate studies from our lab, that are partly published or available as preprints (Song et al., *JCB* 2018 and Thawani et al., *eLife* 2019). Visualization of recombinant GFP-augmin and γ -TuRC via a fluorescent primary antibody seems to show that these proteins are localized rather specifically to branch-points. Most importantly, augmin/ γ TuRC binding occurs after TPX2 localized to MTs in extract, which was initially a surprising finding and highlights the importance of TPX2 and its LLPS scaffold on MTs. We address this point in the discussion (pg. 21 paragraph 1).

What may be more relevant to this manuscript is that the *in vitro* reconstitutions from our lab (Alfaro-Aco et al., 2019 bioRxiv) show that there is a benefit in having TPX2 bind to the MT first. Alfaro-Aco, et al., show that TPX2 helps recruit augmin to the MT lattice and that γ -TuRC accumulates on TPX2 condensate patches, as shown by negative stain EM below.

In sum, these separate studies highlight the importance of TPX2 and its co-condensation on the MT as an origin of branching MT nucleation. In the discussion (pg's. 20 and 21) we incorporated the aforementioned points on the distribution of branch sites and the role of other factors relative to TPX2, and how these findings relate to advances in this MS. We thank the reviewer again for these insightful questions that improved this manuscript.

Below are the data from previous publications that visualized augmin, TPX2 and γ -TuRC on branched MT networks, as a reference.

Augmin (a) Song et al., *JCB* 2018

(b) Thawani et al., *eLife* 2019

γ -TuRC:
Thawani et al., *eLife* 2019

E Observing γ -TuRC in branched networks

XMAP215:

Unpublished data – Matthew King (IF of XMAP215 on fixed branched networks)

Minor concerns

1) Consider replacing the term 'mono-disperse' with 'non-phase-separated'.

We agree with the reviewer and have changed 'mono-disperse' for 'non-phase-separated' throughout the manuscript.

2) Figure 1A: Difficult to distinguish between low disorder and predicted alpha-helices.

To address this concern we have made several modification to Fig. 1A. The cartoon alpha-helices have been enlarged so that they can be more easily distinguished and they are now connected by a line (which denotes no predicted secondary structure).

3) Figure 1B: Consider a better DIC image.

We thank the reviewer for this suggestion. We chose this image because it directly compares GFP-TPX2 to untagged TPX2 - both images having been taken on the same day, using the same microscope and the same concentration for both proteins. Unfortunately, the heterogeneity in condensate size (which are 3-D spheres) prevents ideal focusing and creates a shadowing effect using DIC microscopy - i.e. when large condensates are in focus, small one may not be (below the focal plain). We note that despite these inherent limitations condensate abundance, sizes, and distribution of sizes are similar between fluorescent and untagged condensates. This result gave us confidence that the GFP tag did not affect the LLPS nature of TPX2 and the LLPS is an inherent feature of TPX2.

4) Define 'partition co-efficient' in the figure itself.

We thank the reviewer for this suggestion and we have now included a graphical illustration in Figure 1C and additional lines of text (results section pg. 5, paragraph 2) that explain partition coefficient.

5) Figure 3D and 4C: Why are the 'full-length' data so different? Have these been accidentally misrepresented?

We thank the reviewer for noticing this difference in the fit curve between Figure 3D and 4C. In fact, both curves are identical, except for the fact that the 4C line is half the width of the 3D line and the points (circles) have been removed. These modifications were made to de-emphasize the full length data in 4C, primarily so that the other fit curves (of the chimera constructs), which fall on top of this curve, are more visible and because full length line is there as reference for the chimera data. An identical procedure was carried out for the CT_480-716 data.

6) The mechanism of how importin alpha/beta inhibits phase separation of TPX-2/tubulin should be discussed – is this a competitive binding mechanism? Are there other examples of such mechanism in phase separation regulation?

We thank the reviewer for pointing out this oversight. We incorporated several sentences in the discussion section that pertain to the possible mechanism of importin inhibition of TPX2-tubulin co-condensation. Based on literature of karyopherins disrupting phase separation and structural characterizations of TPX2, we claim that inhibition is not likely due to competitive binding. Rather it seems that since phase separation of TPX2 involves most of the polypeptide sequence, importin inhibition, too, is achieved by disrupting multiple distributed throughout TPX2. In fact, now that we know that importins inhibit LLPS of TPX2, this warrants its own study on how this works mechanistically. Also, we are very interested in how phase transitions are inhibited and are following this up in future studies.

7) The abstract should be revised to remove claims not supported by current data.

We thank the reviewer for this suggestion. We have modified our language in the abstract to be more true to the data, given that we made substantial additions to the manuscript thanks to the reviewer.

Works cited:

Alfaro-Aco, R., Thawani, A., and Petry, S. (2019). Biochemical reconstitution of branching microtubule nucleation. *BioRxiv* 700047.

Gayek, A.S., and Ohi, R. (2014). Kinetochore-microtubule stability governs the metaphase requirement for Eg5. *MBoC* 25, 2051–2060.

Mead-Hunter, R., King, A.J.C., and Mullins, B.J. (2012). Plateau Rayleigh Instability Simulation. *Langmuir* 28, 6731-6735.

Reid, T.A., Schuster, B.M., Mann, B.J., Balchand, S.K., Plooster, M., McClellan, M., Coombes, C.E., Wadsworth, P., and Gardner, M.K. (2016). Suppression of microtubule assembly kinetics by the mitotic protein TPX2. *Journal of Cell Science* 129, 1319.

Song, J.-G., King, M.R., Zhang, R., Kadzik, R.S., Thawani, A., and Petry, S. (2018). Mechanism of how augmin directly targets the γ -tubulin ring complex to microtubules. *J Cell Biol* jcb.201711090.

Thawani, A., Kadzik, R.S., and Petry, S. (2018). XMAP215 is a microtubule nucleation factor that functions synergistically with the γ -tubulin ring complex. *Nat. Cell Biol.* 20, 575–585.

Thawani, A., Stone, H.A., Shaevitz, J.W., and Petry, S. (2019). Spatiotemporal organization of branched microtubule networks. *ELife* 8, e43890.

Zhang, R., Roostalu, J., Surrey, T., and Nogales, E. (2017). Structural insight into TPX2-stimulated microtubule assembly. *eLife Sciences* 6, e30959.

REVIEWERS' COMMENTS:

Reviewer #2 (Remarks to the Author):

In the revised version of the manuscript, the authors have adequately addressed by concerns. I recommend publication.

Reviewer #3 (Remarks to the Author):

The authors have improved the manuscript by providing new results and some text editing. Overall, this is a strong manuscript but I am concerned that some claims continue to be inaccurate and/or overstated. The most obvious misrepresentation is the claim that the authors have established direct casual relationship between TPX2 properties determined in the assay with purified proteins and the MT branching, which takes place only in cytosol. E.g. mutant versions of TPX2 are examined for their ability to form co-condensates in vitro using system containing only two proteins (TPX2 and tubulin). In this system MT branches do not form, because important factors are missing. The authors draw their conclusions about MT nucleation using cytosol. Because the authors cannot exclude that these molecular perturbations affect other TPX2 properties, such as interactions with other proteins, these observations are just the suggestive correlations. It is also inappropriate to base conclusions for

these experiments on data in other unpublished work. The wording in this MS should be adjusted to acknowledge that other properties of TPX2 (in addition to its ability to form co-condensates) might be affected and that they may reduce nucleation. Specific examples:

a) In Discussion: "Our results demonstrate that phase separation of TPX2 and tubulin improves the efficiency of branching MT nucleation in cytosol at least 10-fold." This statement assumes that the ability to form co-condensates is directly (and quantitatively) related to nucleation. This has not been established by the authors and is merely suggested by the observed correlations. This statement should be modified to acknowledge this gap in knowledge. Also, to improve readability, please include reference to specific figures that show 10-fold differences in condensate formation and nucleation.

b) In Discussion: "The fact that TPX2 selectively phase separates onto a pre-existing MT enables the autocatalytic amplification of MTs via branching MT nucleation, which increases MT nucleation rates up to 100-fold". Same objection as in a) above. The authors have established that "TPX2 selectively phase separates onto .. MT", so I agree that this is now a fact. However, the quantitative statement that this specific property "enables the autocatalytic amplification of MTs via branching MT nucleation, which increases MT nucleation rates up to 100-fold", as far as I understand, is based on correlative observations using chimeric constructs. As pointed above, the authors have not ruled out

other possible issues with these mutant proteins (reduced binding to other protein factors, etc). Please modify. Also, provide references to figures that show 100-fold effect.

c) MS text: “data suggest that previous observations of TPX2 and tubulin “clusters” / “aggregates” / “puncta” were, in fact, TPX2-tubulin co-condensates that had undergone LLPS”. The authors created oxymoron. This is acceptable as a figure of speech but not for a scientific document. Either the authors have documented a “fact”, or their data merely “suggest”. I believe that the latter description is more accurate, because the authors have not presented evidence for the “fact” they claim (that in previous studies “TPX2-tubulin co-condensates ... had undergone LLPS”).

Other major issues:

2. The paper is significantly improved by adding FRAP measurements for TPX2 but analogous observation for tubulin is “not shown” (Fig S4B). TPX2 exchanges very slowly (see below), does tubulin have similar mobility within the co-condensates? This information is important for improving our understanding of how exactly TPX2 promotes tubulin nucleation. FRAP measurements for tubulin should be provided.

3. Video 4 provides very important data that are central for this paper: “GFP_TPX2 and Cy5-tubulin co-condensates on a stabilized MT in vitro, imaged over time.” However, “only GFP (GFP_TPX2) channel is shown”. Please include tubulin channel.

4. No IDR-TB and IDR-noTB co-condensates show equally low level of tubulin enrichment (Fig. 5B) but they are described as “normal” tubulin binding (green check mark) “no” tubulin binding (red cross) in Fig. 5D. This requires clarification.

5. I am concerned about the use of “partition coefficient” to quantify MT decoration by TPX2. The authors define “Partition coefficient .. as the difference in mean intensity of a condensate compared to background, i.e. apparent relative enrichment” (Page 27). However, they use this terminology to describe results of the experiments carried out at very low TPX2 concentration (Fig. 2 E, G subnanomolar and nanomolar), when there is no evidence that TPX2 has formed condensates. In fact, since the authors conclude that condensates on MTs are formed similarly to those in solution, it is very likely that at these low concentrations, the MT-bound TPX2 is non-phase separated. The Y-axis label on these graphs is therefore misleading and should be replaced with the more accurate one (normalized brightness, etc).

6. The authors have added new results from FRAP experiments but a thoughtful discussion of these results is missing. The data show very little exchange of TPX2 within the droplets. It is very slow and recovers to only 30%, indicating that MOST of the TPX2 is stable. As noted very briefly in the MS by the authors (Fig S2), there is some “aging” of these coacervates, so I am concerned that TPX2

mobility might be slowing down with their age. The authors have not ruled out that with time, the coacervates harden and TPX2 becomes even more ordered. The authors also observed heterogeneity of co-condensates with respect to their ability to nucleate MTs. These features (the exchange rate, aging and functional behavior) may or may not be related, and this aspect should be investigated in the future. The current study should pose these questions and provide baseline observations for future work. Thus, a brief discussion of the kinetic changes of co-condensates' properties and the issue

with slow FRAP recovery should be discussed.

Minor comments

1) Figure 1 – scale bars size is missing

2) Figure 5: this approach uses specific proteins to create chimeras, but these proteins are not introduced and references are not provided. Please add references and a brief description of these proteins to explain their choice for Fig 5 (FUS, etc)

3) In Methods: All proteins used in this study should be listed with identifying numbers (e.g. NCBI). Current description is not sufficient (“FUS and Ran are the human versions”).

4) Fig 7 model has one inaccuracy, which is easy to correct. Both cartoons show even distribution of proteins along the MT. According to this MS, the cartoon on the right should show formation of “puncta” rather than even coating at high TPX2 concentration.

5) Page 10: “..TPX2 displays similar recovery kinetics on MTs (Fig. S4B) as it does as a condensate in solution (Fig. 1C)”. Wrong figure is cited, should be S1C

6) X-axis label for plots in Fig2 panels E and G and similar is misleading. As far as I understand, these experiments were done with equimolar concentrations of two proteins, but only TPX2 is listed in the X-axis labels.

7) “Our new data (Fig 2 H-I and S4) indicate that the material state of TPX2 on the MT is similar to that in solution”. Terminology (“material state”) should be explained or not used.

Reviewer 3 Comments

Authors' response in red

The authors have improved the manuscript by providing new results and some text editing. Overall, this is a strong manuscript but I am concerned that some claims continue to be inaccurate and/or overstated. The most obvious misrepresentation is the claim that the authors have established direct casual relationship between TPX2 properties determined in the assay with purified proteins and the MT branching, which takes place only in cytosol. E.g. mutant versions of TPX2 are examined for their ability to form co-condensates *in vitro* using system containing only two proteins (TPX2 and tubulin). In this system MT branches do not form, because important factors are missing. The authors draw their conclusions about MT nucleation using cytosol. Because the authors cannot exclude that these molecular perturbations affect other TPX2 properties, such as interactions with other proteins, these observations are just the suggestive correlations. It is also inappropriate to base conclusions for these experiments on data in other unpublished work. The wording in this MS should be adjusted to acknowledge that other properties of TPX2 (in addition to its ability to form co-condensates) might be affected and that they may reduce nucleation.

We thank the reviewer for their complement about the strength of the manuscript and their concern regarding some of the claims, which were helpful to perfect this manuscript. We agree that our phrasing in the original manuscript incorrectly implied the elucidation of a “direct causal relationship” between our *in vitro* and in cytosol data. Below we detail the many modifications to our language to avoid this overstatement and ensure that this relationship is understood as suggestive. In addition, we acknowledge other properties of TPX2 that may be affected in these experiments and in TPX2's activity. Overall, we believe that we have thoroughly addressed the reviewer's points to substantially improve this manuscript.

Specific examples:

a) In Discussion: “Our results demonstrate that phase separation of TPX2 and tubulin improves the efficiency of branching MT nucleation in cytosol at least 10-fold.” This statement assumes that the ability to form co-condensates is directly (and quantitatively) related to nucleation. This has not been established by the authors and is merely suggested by the observed correlations. This statement should be modified to acknowledge this gap in knowledge. Also, to improve readability, please include reference to specific figures that show 10-fold differences in condensate formation and nucleation.

We thank the reviewer for pointing out this specific oversight. We rewrote the sentence to read as follows (changes *italicized*):

“We observe a correlation between TPX2-tubulin co-condensation and branching MT nucleation efficiency in the physiological context of *Xenopus* cytosol (Figures 4 and 5). These results *suggest* that phase separation of TPX2 and tubulin *could underlie* the 10-fold improvement in the branching MT nucleation efficiency (*Figure 4*).”

We believe these changes appropriately reframe our results as suggestive, but not conclusive. Additionally, please note that the appropriate figures for the 10-fold effect have been referenced. Please refer to page 15, lines 749-750.

b) In Discussion: “The fact that TPX2 selectively phase separates onto a pre-existing MT enables the autocatalytic amplification of MTs via branching MT nucleation, which increases MT nucleation rates up to 100-fold”. Same objection as in a) above. The authors have established that “TPX2 selectively phase separates onto .. MT”, so I agree that this is now a fact. However, the quantitative statement that this specific property “enables the autocatalytic amplification of MTs via branching MT nucleation, which increases MT nucleation rates up to 100-fold”, as far as I understand, is based on correlative observations using chimeric constructs. As pointed above, the authors have not ruled out other possible issues with these mutant proteins (reduced binding to other protein factors, etc). Please modify. Also, provide references to figures that show 100-fold effect.

We thank the review for highlighting our incorrect combination of these two ideas and the resulting over-statement. We decoupled the two ideas and the text has been edited as follows (Page: 15 lines 752-756):

“Our data suggests that the specificity of TPX2 to promote branching MT nucleation could result from preferential phase separation onto a pre-existing microtubule. Lastly, TPX2-mediated branching MT nucleation increases MT nucleation rates up to 100-fold²² (Table 1), in an all-or-none manner, at a concentration of TPX2 that matches the phase boundary of TPX2-tubulin co-condensation.”

We believe this provides the requested modification to isolate the quantitative effect of TPX2 to cause a 100-fold increase from TPX2’s localization to a MT.

c) MS text: “data suggest that previous observations of TPX2 and tubulin “clusters” / “aggregates”/ “puncta” were, in fact, TPX2-tubulin co-condensates that had undergone LLPS”. The authors created oxymoron. This is acceptable as a figure of speech but not for a scientific document. Either the authors have documented a “fact”, or their data merely “suggest”. I believe that the latter description is more accurate, because the authors have not presented evidence for the “fact” they claim (that in previous studies “TPX2-tubulin co-condensates ... had undergone LLPS”).

We thank the review for pointing out our oversight in using colloquial language. We agree that our data is merely suggestive and modified the sentence to read as follows (changes *italicized*): “These data suggest that previous observations of TPX2 and tubulin clusters, aggregates, and puncta *could have been* manifestations of TPX2-tubulin co-condensation via LLPS.” This sentence can be found on page 5, lines 99-100.

Other major issues:

2. The paper is significantly improved by adding FRAP measurements for TPX2 but analogous observation for tubulin is “not shown” (Fig S4B). TPX2 exchanges very slowly (see below),

does tubulin have similar mobility within the co-condensates? This information is important for improving our understanding of how exactly TPX2 promotes tubulin nucleation. FRAP measurements for tubulin should be provided.

We thank the reviewer for this excellent suggestion. We have included additional data in Supplementary Figure 4 panels D-F, which show the FRAP profile of Cy5-tubulin co-condensed with TPX2 on a MT. We find that the recovery kinetics of tubulin are similar to TPX2. We report this observation in the main text on pages 7-8 lines 230-239.

3. Video 4 provides very important data that are central for this paper: “GFP_TPX2 and Cy5-tubulin co-condensates on a stabilized MT *in vitro*, imaged over time.” However, “only GFP (GFP_TPX2) channel is shown”. Please include tubulin channel.

We thank the reviewer for pointing out this oversight. We updated Movie 4 to include the Cy5-Tubulin and Alexa568-MT channels, as well as a merged channel.

4. No IDR-TB and IDR-noTB co-condensates show equally low level of tubulin enrichment (Fig. 5B) but they are described as “normal” tubulin binding (green check mark) “no” tubulin binding (red cross) in Fig. 5D. This requires clarification.

We thank the review for this important observation. We included the following sentence to clarify this aspect: “Interestingly, the NoIDR_TB chimera only showed a slight improvement in tubulin co-partitioning relative to the IDR_NoTB *in vitro*, whereas the TOG domains used here have been well characterized to interact with tubulin in *Xenopus* egg cytosol^{19,40}.” This can be found on page 10 lines 494-497. Additionally, we removed the ‘Phase Separation’ and ‘Tubulin Binding’ columns from the summary table, to prevent ambiguity about those definitions. The resulting summary table is more concise and easier to understand.

To elaborate on our clarification, we note that TOG domains used in the No-IDR-TB chimeras have previously been characterized to interact with tubulin *in vitro* and functionally in *Xenopus* cytosol (Brouhard 2008, Weber 2013, and Thawani 2018 – full citations below). Furthermore, there are likely differences in the stoichiometries and manner in which the endogenous TPX2 N-terminal region (disordered) vs. the TOG1-2 domains (structured) interact with tubulin, which may underlie the difference in the tubulin partition coefficient. Last, the TOG-tubulin interaction is salt-dependent, and with less salt more ‘structured’ tubulin binding via TOG domains can be achieved. Our lab is quite interested in understanding the biophysical basis of co-condensation on the microtubule lattice, as opposed to structured interactions e.g. by molecular motors, and this topic is being further investigated in our next generation of papers. Nonetheless, we believe we adequately addressed the ambiguity about our TOG-chimera – tubulin interaction via modifications to the text and the summary table.

Brouhard, Gary J., Jeffrey H. Stear, Tim L. Noetzel, Jawdat Al-Bassam, Kazuhisa Kinoshita, Stephen C. Harrison, Jonathon Howard, and Anthony A. Hyman. “XMAP215 Is a Processive Microtubule Polymerase.” *Cell* 132, no. 1 (January 11, 2008): 79–88.
<https://doi.org/10.1016/j.cell.2007.11.043>.

Reber, Simone B., Johannes Baumgart, Per O. Widlund, Andrei Pozniakovsky, Jonathon Howard, Anthony A. Hyman, and Frank Jülicher. “XMAP215 Activity Sets Spindle Length by

Controlling the Total Mass of Spindle Microtubules.” *Nature Cell Biology* 15, no. 9 (September 2013): 1116. <https://doi.org/10.1038/ncb2834>.

Thawani, Akanksha, Rachel S. Kadzik, and Sabine Petry. “XMAP215 Is a Microtubule Nucleation Factor That Functions Synergistically with the γ -Tubulin Ring Complex.” *Nature Cell Biology* 20, no. 5 (May 2018): 575–85. <https://doi.org/10.1038/s41556-018-0091-6>.

5. I am concerned about the use of “partition coefficient” to quantify MT decoration by TPX2. The authors define “Partition coefficient .. as the difference in mean intensity of a condensate compared to background, i.e. apparent relative enrichment” (Page 27). However, they use this terminology to describe results of the experiments carried out at very low TPX2 concentration (Fig. 2 E, G sub nanomolar and nanomolar), when there is no evidence that TPX2 has formed condensates. In fact, since the authors conclude that condensates on MTs are formed similarly to those in solution, it is very likely that at these low concentrations, the MT-bound TPX2 is non-phase separated. The Y-axes label on these graphs is therefore misleading and should be replaced with the more accurate one (normalized brightness, etc).

We thank the reviewer for this important observation. We modified the y-axes to reflect “Relative fluorescence - Normalized to max”. We hope this clarifies that these measurements do not exclusively reflect phase separation, but also non-phase separated localization. Please find these modifications in Figure 2.

6. The authors have added new results from FRAP experiments, but a thoughtful discussion of these results is missing. The data show very little exchange of TPX2 within the droplets. It is very slow and recovers to only 30%, indicating that MOST of the TPX2 is stable. As noted very briefly in the MS by the authors (Fig S2), there is some “aging” of these coacervates, so I am concerned that TPX2 mobility might be slowing down with their age. The authors have not ruled out that with time, the coacervates harden and TPX2 becomes even more ordered. The authors also observed heterogeneity of co-condensates with respect to their ability to nucleate MTs. These features (the exchange rate, aging and functional behavior) may or may not be related, and this aspect should be investigated in the future. The current study should pose these questions and provide baseline observations for future work. Thus, a brief discussion of the kinetic changes of co-condensates’ properties and the issue with slow FRAP recovery should be discussed.

We thank the reviewer for this thoughtful suggestion. We discuss all of these points and raise questions for future research in the edited manuscript. These can be found on pages 12-13 lines 602-656.

Minor comments

1) Figure 1 – scale bars size is missing

Thank you for pointing out this mistake. The scale bar sizes were added to the legend.

2) Figure 5: this approach uses specific proteins to create chimeras, but these proteins are not introduced, and references are not provided. Please add references and a brief description of these proteins to explain their choice for Fig 5 (FUS, etc.)

We thank the reviewer for pointing out this oversight. We included several additional sentences and appropriate citations to clarify why these protein domains were used in generating the chimera constructs on page 10 lines 485-489, and as cited below:

“First, the intrinsically disordered N-terminal region of Fused in Sarcoma (FUS) was used to replace TPX2’s N-terminal part, since it readily phase separates³⁸ but is not expected to associate with tubulin owing to its negative pI (IDR_NoTB, Fig. 5A). Conversely, TOG domains 1 and 2 from XMAP215 were used, because they are well structured and not expected to contribute to phase separation, but validated to functionally interact with tubulin^{19,39,40} (NoIDR_TB, Fig. 5A).”

3) In Methods: All proteins used in this study should be listed with identifying numbers (e.g. NCBI). Current description is not sufficient (“FUS and Ran are the human versions”).

We thank the reviewer for this important point. NCBI GeneID #'s are now included as identifiers for each protein (please refer to page 16).

4) Fig 7 model has one inaccuracy, which is easy to correct. Both cartoons show even distribution of proteins along the MT. According to this MS, the cartoon on the right should show formation of “puncta” rather than even coating at high TPX2 concentration.

We thank the review for this suggestion, which has been corrected in Figure 7.

5) Page 10: “.TPX2 displays similar recovery kinetics on MTs (Fig. S4B) as it does as a condensate in solution (Fig. 1C)”. Wrong figure is cited, should be S1C

We thank the reviewer for pointing out this error. It has been corrected and can be found on page 8, line 241.

6) X-axis label for plots in Fig2 panels E and G and similar is misleading. As far as I understand, these experiments were done with equimolar concentrations of two proteins, but only TPX2 is listed in the X-axes labels.

We thank the reviewer for this observation. We updated the label on the X-axis to include tubulin in figure 2

7) “Our new data (Fig 2 H-I and S4) indicate that the material state of TPX2 on the MT is similar to that in solution”. Terminology (“material state”) should be explained or not used.

This term was used only in the rebuttal to address a point made by reviewer 1. We agree with the reviewer that the terminology ‘material state’ is jargon, and we therefore refrained from using it in the manuscript.